- A Study of the Dependence between Soil Moisture and
- 2 Precipitation in different Ecoregions of the Northern
- 3 Hemisphere

Shouye Xue<sup>a</sup> and Guocan Wu <sup>a\*</sup>

- <sup>a</sup> State Key Laboratory of Earth Surface Processes and Disaster Risk Reduction,
- 8 Faculty of Geographical Science, Beijing Normal University, Beijing, 100875, China


\*Corresponding author: Guocan Wu, gcwu@bnu.edu.cn


## Abstract






























Soil moisture plays a critical role in the land-atmosphere coupling system. It is replenished by precipitation and transported back to the atmosphere through land surface evaporation and vegetation transpiration. Soil moisture is, therefore, influenced by both precipitation and evapotranspiration, with spatial heterogeneities and seasonal variations across different ecological zones. The relationship between soil moisture and precipitation was found to be nonlinear and negative in Northern Hemisphere ecosystems. However, the driving mechanisms of these negative correlations, especially how soil moisture is influenced by precipitation and evapotranspiration, still remain unclear. This study quantified the spatiotemporal distribution of the nonlinear dependence of soil moisture to precipitation, and identify the dominant factors in different ecoregions to explore the driving mechanisms and regional patterns. The joint distributions of precipitation and soil moisture were analyzed at monthly and annual scales, using soil moisture and precipitation data from ERA5-Land and Global Precipitation Climatology Project, respectively. The nonlinear negative dependences reached to 19.2%, 0.7%, and 2.3% at monthly scale, while were 3.0%, 4.0%, and 8.6% at annual scale, respectively, for the three soil layers. These negative dependences were shown to be most prominent in temperate grasslands, savannas, shrublands, deserts, xeric shrublands, and tundra regions, where driven by the land surface temperature and by the air temperature–gross primary production relationship at the monthly scale based on Ridge regression models and Bayesian models. Additionally, the negative dependence is also linked to freeze-thaw cycles, precipitation seasonality, and temperature fluctuations, which lead to asynchronous changes between soil moisture and precipitation at the seasonal scale. At the annual scale, the negative dependence was associated with long-term changes in precipitation and temperature that affect vegetation and surface properties, by altering soil water capacity. These findings enhance the understanding of land-atmosphere interactions providing a valuable basis for future research on drought, hydrometeorology, and ecological conservation.

Keywords: climate change, precipitation, soil moisture, ecoregions

## 1. Introduction






























Soil moisture is a critical source of water for vegetation growth, replenished by precipitation and groundwater, and returned to the atmosphere through evapotranspiration. It plays a key role in weather conditions, vegetation dynamics, and groundwater storage (Li et al., 2022; Qiao et al., 2023; Vereecken et al., 2008; Zhou et al., 2021), with significant implications for the global climate. Surface soil moisture regulates the distribution of available energy at the land surface and exchanges energy with the near-surface atmosphere through sensible and latent heat fluxes, thereby controlling the surface energy balance (Haghighi et al., 2018; McColl et al., 2017). In contrast, deep soil moisture is more directly influenced by vegetation growth, particularly by the development of plant roots, which play a crucial role in the vertical infiltration of precipitation into deeper soil layers (Szutu and Papuga, 2019; Xiao et al., 2024; Xue and Wu, 2024). Precipitation variability, which refers to the amplitude of precipitation fluctuations over different times, influences soil moisture and thereby land surface coupling (Koster et al., 2009; Taylor et al., 2012). Precipitation patterns are reported to have undergone significant changes in recent decades (Lv et al., 2023; Mao et al., 2022; Wu et al., 2021), mainly manifested as anthropogenic amplification of precipitation variability (Zhang et al., 2024). The increase in the frequency of extreme precipitation events (Myhre et al., 2019; Wang et al., 2022) and decrease in the frequency of smaller precipitation events (Ma et al., 2015) amplify soil moisture fluctuations and prolong the moisture stress periods between consecutive precipitation events (Knapp et al., 2008). This can directly affect vegetation growth and soil moisture responses (Feldman et al., 2024; He et al., 2023), particularly through changes in the duration and intensity of soil evaporation and plant transpiration (Gu et al., 2021; Wullschleger and Hanson, 2006). Soil moisture has been shown to be negatively correlated with precipitation in certain regions, based on Pearson correlation analyses (Cook et al., 2006; Yang et al., 2018). The changes in soil moisture at different depths also show notable discrepancies (Shen et al., 2016; Zhu et al., 2014). Surface soil moisture has been shown to respond to precipitation

approximately a month earlier than deeper soil moisture, with a more pronounced positive correlation between precipitation and soil moisture occurring at depths greater than 50 cm (Zhang et al., 2020).






























Most current analyses of the relationship between soil moisture and precipitation assume a linear relationship (Sehler et al., 2019; Yang et al., 2018). In reality, the response of soil moisture to precipitation is extremely complex and often nonlinear (Drager et al., 2022). This kind of nonlinear and asymmetric correlation is generally referred to as "dependence". Existing studies have not fully addressed some issues in the nonlinear dependence of soil moisture to precipitation, including the heterogeneity in different ecoregions and soil layers, as well as inadequate identification of tail dependence. Moreover, the factors driving this negative dependence between soil moisture and precipitation remain poorly understood due to the complicated land atmosphere coupling processes, particularly in the Northern Hemisphere where different types of vegetation coverage are present. Among the methods used to explore nonlinear relationships, the copula function is one of the most widely applied approaches for modeling the joint distributions of precipitation and soil moisture (Cammalleri et al., 2024). The copula is a stochastic model that can reveal nonlinear and asymmetric dependence structures, which are difficult to capture using traditional linear methods. It provides a flexible framework for modeling joint distributions of multiple variables, allowing for a more precise understanding of the evolving dependence of soil moisture on precipitation than that offered by traditional linear regression and correlation methods.

In terms of the water cycle, soil moisture is replenished by precipitation and groundwater, while also being absorbed by plant roots and lost through evapotranspiration. Therefore, the change of soil moisture is actually simultaneously influenced by precipitation volume, frequency, and evapotranspiration. However, the response of soil moisture to precipitation and evapotranspiration varies across different time scales, presented as nonlinear and asymmetric. The long-term effects of changes in evapotranspiration and precipitation on soil moisture are further shaped by seasonal

transitions, with significant differences observed at different soil depths (Szutu and Papuga, 2019). These differences are influenced by factors such as soil freeze—thaw processes and vegetation community structure. Therefore, the relative contributions of evapotranspiration, precipitation volume, and frequency to soil moisture changes should be quantified at different time scales.

Although previous studies have identified the mechanisms of soil moisture variation across different time scales (Shen et al., 2018; Vidana Gamage et al., 2020), the interaction among precipitation, evapotranspiration and soil water under climate change may have changed over different time scales. In particular, although the negative dependence has been reported, its dominant drivers and their relative contributions across different timescales and soil layers still remain unclear. The dependence of soil moisture to precipitation and its interactions with evapotranspiration under conditions of climate change require further investigation. Accordingly, the ridge regression models for precipitation amount, precipitation frequency, evapotranspiration, and soil moisture can be used to quantify the relative influence of precipitation and evapotranspiration on soil moisture. As an improvement of the least squares estimation method, it can handle the multi-collinearity problems of the covariates, although it is usually biased.

This study targets the nonlinear dependence of soil moisture to precipitation across Northern Hemisphere at monthly and annual scales from 2000 to 2019. A copula function was applied to describe the joint distribution of precipitation and soil moisture. It can capture the asymmetric and tail-dependent relationship, as well as the varying influences of precipitation volume, frequency, and evapotranspiration on soil moisture at monthly and seasonal scales. A Bayesian attribution framework involved gross primary productivity (GPP), land surface temperature (LST), and near-surface air temperature (T<sub>a</sub>) were selected to identify the key driving factors, since the dependence between precipitation and soil moisture is influenced by factors such as vegetation growth, temperature, and soil properties. The driving factors and regional characteristics of the negative correlation observed between precipitation and soil

moisture in different ecoregions were also compared. This study enhances the understanding of complex interactions between key meteorological factors such as precipitation, evapotranspiration, and soil moisture under climate change, and provides a basis for future land–atmosphere coupling system modeling.

# 2. Material and Method

### 2.1 Material

#### 2.1.1 Soil moisture

The soil moisture data used in this study were obtained from the fifth generation of reanalysis from the European Centre for Medium-Range Weather Forecasts (ECMWF), using atmospheric forcing to control the simulated land field variables and provide the land components (ERA5-Land) (Muñoz Sabater, 2019). ERA5-Land provides a consistent description of the evolution of the energy and water cycles over land, and therefore, has been widely used in various land surface applications such as flood or drought forecasting (Joaquín Muñoz-Sabater, 2021). The ERA5-Land soil moisture data are available for four layers, 0 to 7, 7 to 28, 28 to 100, and 100 to 289 cm, at a  $0.1^{\circ} \times 0.1^{\circ}$  spatial and hourly temporal resolution from 1950 to present. The soil moisture from the first three soil layers during 2000 to 2019 were used. They were resampled to a  $0.25^{\circ} \times 0.25^{\circ}$  spatial resolution and averaged to daily, monthly, and yearly scales to be consistent with other variables in this study.

## 2.1.2 Precipitation

The Global Precipitation Climatology Project (GPCP) is a global precipitation project that integrates infrared and microwave data from multiple geostationary and polar-orbiting satellites, and corrected by many meteorological station observations (Adler et al., 2003; Huffman and Bolvin, 2013). It is an important component of the Global Energy and Water Cycle Experiment (GEWEX) in the World Climate Research Programme (WCRP). A daily precipitation field with a 1° × 1° resolution since 1996 was generated by integrating the satellite products and then adjusting the daily

precipitation by monthly data observed from the ground to make it consistent with the meteorological observations. Daily precipitation was resampled to a  $0.25^{\circ} \times 0.25^{\circ}$  spatial resolution and then used to calculate the total precipitation volume and precipitation frequency at the monthly, seasonal, and annual scale from 2000 to 2019.

### 2.1.3 Covariate variables

## 2.1.3.1 Gross primary production

The gross primary production (GPP) dataset was from the Vegetation Optical Depth Climate Archive v2, which used microwave remote sensing estimates of vegetation optical depth to estimate the GPP at the global scale for the period 1988 to 2020 (Wild et al., 2022). These GPP data were trained and evaluated against FLUXNET in-situ observations and compared with largely independent state-of-the-art GPP datasets from the Moderate Resolution Imaging Spectroradiometer (MODIS). The Vegetation Optical Depth Climate Archive v2 GPP dataset has a  $0.25^{\circ} \times 0.25^{\circ}$  spatial and half-monthly temporal resolution, covered from 2000 to 2019.

### 2.1.3.2 Near surface air temperature

The air temperature data ( $T_a$ ) were obtained from the Climatic Research Unit gridded Time Series (CRU TS), which is one of the most widely used climate datasets and is produced by the National Centre for Atmospheric Sciences in the United Kingdom. CRU TS v4.07 was derived by the interpolation of monthly climate anomalies from extensive networks of weather station observations (Harris et al., 2020). It provides monthly land surface data from 1901 to 2020 at a  $0.5^{\circ} \times 0.5^{\circ}$  resolution worldwide. The mean temperatures at the monthly, seasonal, and annual scales during 2000 to 2019 were calculated and resampled to a  $0.25^{\circ} \times 0.25^{\circ}$  spatial resolution.

#### 2.1.3.3 Land surface temperature

Land surface temperature (LST) data were accessed from the European Space Agency Climate Change Initiative (CCI), which is funded by the European Space Agency as part of the Agency's CCI Program. It aims to significantly improve current satellite LST data records to meet the challenging Global Climate Observing System requirements for climate applications and realize the full potential of long-term LST

data for climate science (Hollmann et al., 2013). These data were the first global LST climate data records of over 25 years at a  $0.25^{\circ} \times 0.25^{\circ}$  resolution and with an expected error within 1 K. The LST dataset included ascending and descending orbit data, which were used to calculate the mean value of separate annual and monthly averages during 2000 to 2019.

#### 2.1.3.4 Evapotranspiration

Evapotranspiration data were accessed from the Global Land Evaporation Amsterdam Model (GLEAM) v3.8a, which provides data of the different components of land evapotranspiration, including transpiration, bare-soil evaporation, interception loss, open-water evaporation, and sublimation, in addition to other related variables such as surface and root-zone soil moisture, sensible heat flux, potential evaporation, and evaporative stress conditions (Miralles et al., 2011). The monthly, seasonal, and annual averages during 2000 to 2019 were calculated based on a 0.25° × 0.25° spatial resolution.

#### 2.1.3.5 Terrestrial ecoregions

Data on terrestrial ecoregions around the globe were accessed from the Conservation Biology Institute (Olson et al., 2001). These ecoregions are relatively large units of land containing distinct assemblages of natural communities and species, with boundaries that approximate the original extent of natural communities prior to major land-use changes. The delineations were completed based on hundreds of previous biogeographical studies and were refined and synthesized using existing information in regional workshops over the course of 10 years to assemble the global dataset (Olson et al., 2001). An ecological layer file encompassing 16 major categories was downloaded.

Although the Köppen climate classification provides a standardized framework based on temperature and precipitation, it may perform not well in accounting for critical biophysical factors, particularly for vegetation. Alternatively, the ecoregion divisions integrate both climatic and ecological factors, offering a more comprehensive understanding of the spatial heterogeneity in vegetation types and hydrological

processes (Gerken et al., 2019; Olson et al., 2001). This makes it particularly advantageous for studying land–atmosphere interactions, since vegetation plays a central role in regulating energy and water fluxes. Therefore, this study adopts ecoregion boundaries to better capture the vegetation related variability in precipitation–soil moisture relationship. Since soil moisture dynamics and their feedbacks with precipitation are strongly influenced by vegetation structure, root systems, and edaphic properties, the ecoregions can provide a more mechanistic and spatially relevant framework for our analysis. All of the T<sub>a</sub>, LST, GPP, soil moisture, and precipitation datasets were masked by these 16 terrestrial ecoregions (Fig. 1) in a 0.25° grid, and monthly, seasonal, or annual mean values in the regions were calculated separately.

Fig. 1 The 16 Terrestrial Ecoregions of the Northern Hemisphere.

### 2.2 Method

#### 2.2.1 Joint distribution

In this study, the joint distribution between precipitation and soil moisture from depths of 0 to 7 cm, 7 to 28 cm, and 28 to 100 cm, using the copula function at both the monthly and annual scales was established. A copula function links multivariate distribution functions with their one-dimensional marginal distributions, and is used for the examination of dependences between multiple variables. It captures nonlinear

dependence structures through joint and marginal probabilities of a pair of variables in complex multivariate systems (Nelsen, 2005). In this study, the copula function was used to explore the nonlinear dependence between precipitation and soil moisture (Equation 1):

238 
$$F_{P,SM}(x,y) = C(F_P(x), F_{SM}(y)), \tag{1}$$

where  $F_P(x)$  and  $F_{SM}(y)$  denote the marginal distribution of precipitation and soil moisture, respectively, and C(u,v) is the copula function linking these two variables. The process for establishing the joint distribution was as follows: (1) The marginal distributions of precipitation and soil moisture were fitted using an automatic optimization function. (2) The most suitable copula function was selected based on the Akaike Information Criterion (AIC) values at the grid level, including Gaussian copula, Student's t copula, Clayton copula, and 37 other copula functions. Different copula functions may be selected for different grid cells. (3) The chosen copula function was then used to compute the corresponding Kendall's tau  $(\tau)$ , upper tail dependence  $(\lambda_U)$ , and lower tail dependence  $(\lambda_L)$ .

The statistic  $\tau$  measures the correlation between two variables to determine the presence of a monotonic relationship.  $\lambda_U$  and  $\lambda_L$  represent the likelihood that, when one variable reaches extreme high or low values, the other variable also reaches extreme values. The calculations of  $\tau$ ,  $\lambda_U$ , and  $\lambda_L$  are based on the dependence parameters of the joint distribution of precipitation and soil moisture, and depends on the selected copula function using the AIC method. Taking the Tawn copula function as an example, the calculation of  $\tau$ ,  $\lambda_U$ , and  $\lambda_L$  are based on the following equations.

$$\tau = 1 - \frac{2\delta}{\theta + 1} + \frac{2\delta^2}{2\theta + 1},\tag{2}$$

257 
$$\lambda_{\rm U} = (1 - \delta) \cdot (2 - 2^{1/\theta}), \tag{3}$$

and

$$\lambda_{L} = \delta \cdot (2 - 2^{1/\theta}), \tag{4}$$

where  $\theta$  is the dependence parameter of the Tawn copula, and  $\delta$  represents the asymmetry parameter. For some copula functions, such as Clayton copula, the Kendall's  $\tau$  values get the priority over the upper and lower tail dependences in the

estimation process. All the calculations were performed using R v4.3.3 with the VineCopula and copula packages, for which detailed calculation methods for  $\tau$ ,  $\lambda_U$ , and  $\lambda_L$  for all copulas are provided. To address the potential delayed response of soil moisture to precipitation, lagged correlation analysis was conducted. For each grid cell, the AIC value was calculated to select copula function (Fig. S1), as shown in the supplementary file. Then the Kendall's tau correlation was calculated between precipitation and soil moisture with time lags ranging from 0 to 12 months (Fig. S2). The lag corresponding to the maximum absolute correlation was identified as the optimal lag.

## 2.2.2 Ridge regression

Ridge regression is designed to address collinear data, although it is a biased estimation method. It is an improved least squares estimation used to generate more reliable regression coefficients at the cost of unbiasedness. Ridge regression outperforms the traditional least squares method when fitting ill-conditioned data (McDonald, 2009). Due to the large uncertainty in precipitation and soil moisture data, ridge regression models were applied for three soil layers, and for both monthly and seasonal scales. Spring was defined as from March to May, summer from June to August, autumn from September to November, and winter from December to February of the following year. Precipitation frequency, volume, and evapotranspiration were treated as predictor variables, with T<sub>a</sub> as a control variable and soil moisture as the response variable.

To clearly differentiate the influence of variables, the regression coefficients for precipitation volume, frequency, and evapotranspiration were normalized using Equation (5) and then assigned to the three primary colors. This approach resulted in a gridded ternary phase diagram.

$$W_i = 1 - \frac{v_i}{\sum_{i=1}^3 v_i},\tag{5}$$

where  $v_i$  ( $v_1$ ,  $v_2$ ,  $v_3$ ) represent precipitation frequency, precipitation volume, and evapotranspiration (ET), respectively, and  $W_i$  refers to the adjusted weight of  $v_i$ .

## 2.2.3 Bayesian generalized non-linear multivariate multilevel models




























The Bayesian generalized non-linear multivariate multilevel model integrates Bayesian inference, generalized linear models, non-linear modeling, multivariate analysis, and hierarchical structures, making it well-suited for complex hierarchical data. It can effectively capture non-linear dependences among multiple response variables (Browne and Draper, 2006; Bürkner, 2017). The model parameters are treated as random variables with prior distributions under the Bayesian framework. Posterior distributions of the parameters are obtained by combining the likelihood function and prior distributions. The Markov Chain Monte Carlo (MCMC) algorithm is then used to resample from the posterior distribution and estimate the posterior means of the parameters to represent the optimal results. Given the hierarchical and multivariate nature of the data, a multilevel structure and multivariate analysis was introduced to model the mixed effects of variables and to capture the relationships among multiple related response variables. Random effects were also incorporated to account for heterogeneity among individuals and reflect the varying effects of univariate or multivariate mixtures on the response variables, thereby improving the accuracy of estimates.

Since the impact approaches of GPP, LST, and T<sub>a</sub> on precipitation (P) and soil moisture (SM) are often unknown, the Gaussian distribution was specified as the prior distribution for these variables in the Bayesian model. To investigate how GPP, LST, and T<sub>a</sub> influence the precipitation–soil moisture coupling relationship, both precipitation and soil moisture were treated as response variables. Bayesian non-linear multivariate multilevel models were developed at both the monthly and seasonal scales, with independent models for 16 ecological zones (Equation 6):

Posterior estimates = 
$$bf(P \sim T_a + GPP + LST + T_a:GPP + T_a:LST + GPP:LST + T_a:GPP:LST) + T_a:GPP:LST + T_a:GP$$

$$bf(SM \sim T_a + GPP + LST + T_a:GPP + T_a:LST + GPP:LST + T_a:GPP:LST), \qquad (6)$$

where the colon represents multivariate mixed effects of different variables; bf stands for Bayesian formula, used to specify each part of the model for P and SM separately; and the "+" combines P and SM into a multivariate model. The model was implemented

in R 4.3.3 using the brms package, which performs diagnostic checks on the sampling results using indicators such as the Gelman–Rubin diagnostic (Rhat statistic) and the effective sample size (ESS). To ensure stability and convergence, four MCMC chains were used for iterative sampling, with each chain running 4,000 iterations, including 2,000 warm-up iterations. A maximum tree depth of 10 was set. Estimate values of all ecoregions were classified into different clusters using the K-means method in R 4.3.3.

## 3. Results

## 3.1 Estimation from the copula function

Fig. 2 Spatial distribution of Kendall's tau ( $\tau$ ), the upper tail dependence ( $\lambda_U$ ), and the lower tail dependence ( $\lambda_L$ ) on the 0.25° × 0.25° grids between monthly precipitation volume and soil moisture during 2000 to 2019. The three columns are for the soil moisture from depths of 0 to 7 cm, 7 to 28 cm, and 28 to 100 cm, respectively.

The copula analysis of monthly average soil moisture and total monthly precipitation volume revealed a clear negative dependence at all three soil depths (Fig. 2(a2, b2, c2)). The percentages of grid cells exhibiting negative dependence at these depths were 19.2%, 0.7%, and 2.3%, respectively. The negative dependence between precipitation and soil moisture is more prevalent in the surface soil layer, where the grid cells exhibiting are more widespread. In contrast, at the middle and deep soil layers,

these negative dependence patterns are primarily confined to the margins of the Sahara desert, the montane grasslands and shrublands, and parts of the deserts and xeric shrublands regions. In the surface layer, the negatively dependent grid patches are more spatially scattered, mainly distributed across the tundra, montane grasslands and shrublands, deserts and xeric shrublands, as well as the tropical and subtropical moist broadleaf forests.

Regions exhibiting high  $\lambda_L$  values were primarily located in the deserts and xeric Shrublands, as well as in parts of India, where  $\lambda_L$  reached values as high as 0.99 (Fig. 2(a1, b1, c1)). With increasing soil depth,  $\lambda_L$  values gradually increased across the Eurasian continent. Similarly,  $\lambda_U$  exhibited a clear reduction in spatial extent with increasing soil depth, with the majority of these regions located in the temperate broadleaf and mixed forests and the southern margin of the Sahara desert. With increasing soil depth,  $\lambda_U$  values consistently decreased, resulting in a lack of clear correspondence between these regions and specific ecological zones (Fig. 2(a3, b3, c3)). This decreasing trend likely reflects the weakening of extreme precipitation—soil moisture coupling in deeper soil layers, except for arid regions where vegetation is sparse or absent.

From the annual scale copula results (Fig. 3), precipitation and soil moisture generally exhibited positive dependences across the entire soil profile. However, negative dependences were observed in regions such as the southern Sahara Desert, Mongolia, and the Elizabeth Islands, reaching 3.0%, 4.0%, and 8.6%, respectively (Fig. 3(a2, b2, c2)). It revealed that the negative correlation was kept between precipitation and soil moisture in long-term scale over arid regions. The negative dependences in these areas expanded outward, primarily concentrated in the montane grasslands and shrublands region. Both the  $\lambda_L$  and the  $\lambda_U$  displayed scattered, patchy distributions, with average values for each soil layer ranging from 0.4 to 0.6.

Fig. 3 Spatial distributions of the  $\tau$ ,  $\lambda_{U}$ , and  $\lambda_{L}$  on the  $0.25^{\circ} \times 0.25^{\circ}$  grids between annual precipitation volume and soil moisture during 2000 to 2019. The three columns are for the soil moisture from depths of 0 to 7 cm, 7 to 28 cm, and 28 to 100 cm, respectively.

# 3.2 Control of soil moisture by precipitation and evapotranspiration

Fig. 4 Ternary map of factors controlling soil moisture, monthly, for the period 2000 to 2019. The bottom-left histogram in the subgraph represents the proportion of grid cells where one variable exerts strong univariate control (with a regression coefficient greater than 75% of the total sum of the three variables), suggesting that soil moisture was predominantly controlled by that specific variable.

On the monthly scale, precipitation exerted the strongest control over soil moisture (Fig. 4), with regions most influenced by precipitation accounting for more than 40% of the variation. These areas were primarily located in the boreal forest/taiga, temperate grasslands, savannas, shrublands, and the eastern part of North America. In contrast,

regions where evapotranspiration predominated were found in Alaska–Northwest Canada, the western United States, the Sahara Desert, and the Middle East. High-latitude regions, especially northern Canada, were primarily influenced by precipitation frequency. Areas where precipitation volume, frequency, and evapotranspiration had similar levels of control were mainly found in Eastern Europe and Russia.





















404

The results from ridge regression revealed more distinct patterns at the seasonal scale compared to the monthly scale (Fig. 5). Soil moisture in spring and summer was mainly controlled by evapotranspiration, which influenced over 40% of grid cells, particularly in the middle soil layers, where it dominated nearly 80%. In contrast, precipitation volume had a greater influence during autumn and winter, particularly in the continental United States, southern Sahara Desert, coastal India, and eastern China. Additionally, as soil depth increased, the influence of evapotranspiration and precipitation frequency gradually intensified. However, in summer, as soil depth increased, the area primarily controlled by precipitation volume expanded (indicated by an increase in the intensity of magenta color in the figures) especially in the eastern United States, Europe, and South Asia. These regions remained strongly influenced by precipitation volume even as evapotranspiration control increased with increasing soil depth during autumn. Northern Russia, Canada, Greenland, and northern Alaska were notably influenced by both precipitation frequency and precipitation volume, with this effect being more pronounced during the non-growing season. In winter, the area controlled by precipitation frequency was larger than that in spring.

Fig. 5 Ternary map of factors controlling soil moisture, seasonally, for the period 2000 to 2019. The bottom-left histogram in the subgraph represents the proportion of the grid cells where one variable exerts strong univariate control (with a regression coefficient greater than 75% of the total sum of the three variables), suggesting that soil moisture was predominantly controlled by that specific variable.

 At the annual scale, precipitation amount exerts a dominant influence across all three soil depth layers, accounting for more than 40% of the total area (Fig. 6). The spatial extent of areas dominated by precipitation amount, precipitation frequency, and evapotranspiration remains largely consistent with that observed at the monthly scale. The regions dominated by precipitation frequency are still primarily located in high-latitude areas, particularly in Greenland and the northern parts of Canada, although no distinct ecological zone patterns are observed in these areas. Regions dominated by precipitation amount are mainly distributed across boreal forests, temperate grasslands, savannas and shrublands, temperate broadleaf and mixed forests, as well as tropical and subtropical moist broadleaf forests. In temperate regions, soil moisture is primarily controlled by precipitation amount due to moderate temperatures and limited rainfall, making substantial precipitation is essential for soil moisture replenishment. In contrast, tropical and subtropical regions experience high temperatures and intense evapotranspiration, requiring substantial precipitation to maintain a water balance.

Fig. 6 Ternary map of factors controlling soil moisture at annual scale, for the period 2000 to 2019. The bottom-left histogram in the subgraph represents the proportion of grid cells where one variable exerts strong univariate control (with a regression coefficient greater than 75% of the total sum of the three variables), suggesting that soil moisture was predominantly controlled by that specific variable.

# 3.3 Drivers of negative dependences between soil moisture and precipitation

For each model in this study, four MCMC chains were used for iterative sampling. The sampling results demonstrated that the chains for both the monthly and annual scales were well-distributed in the parameter space, with no noticeable trends or drifts, indicating convergence to the target posterior distribution. The convergence was considered satisfactory, with all models yielding a Rhat value below 1.05 (Figs. S3, S4).

Fig. 7 Posterior estimates of the covariate variables of the Bayesian generalized non-linear multivariate multilevel model, built using monthly data. The columns represent soil depths of 0 to 7 cm, 7 to 28 cm, and 28 to 100 cm. Red lines indicate linear regressions of precipitation and soil moisture across all ecoregions, with cluster groups represented by three circles. The data point of each ecoregion belongs to a single and non-overlapping cluster.

The negative dependence in the surface layer across the Northern Hemisphere was primarily driven by the interactions between GPP:LST and T<sub>a</sub>:GPP (Fig. 7). It shows that the regression trend line crosses quadrants II and IV. The negative relationship driven by GPP:LST was predominantly concentrated in quadrant IV, where increased precipitation lead to decreased soil moisture in the boreal forest, tundra, temperate coniferous forest, and temperate broadleaf mixed forest. The negative dependence

driven by Ta:GPP was mainly found in quadrant II, with distributions in deserts and xeric shrublands, boreal forests, montane grasslands and shrublands, temperate broadleaf mixed forests, and tundra. For the middle soil layer, GPP:LST drove a negative dependence in tropical and subtropical grasslands, savannas, shrublands, and tropical and subtropical coniferous forests. Ta and Ta:GPP drove in Mediterranean forests, woodlands, and scrub, as well as in temperate grasslands, savannas, and shrublands. The mixed effects of Ta:GPP:LST and Ta:LST had minimal impact across all ecological zones, with all estimates concentrated near the origin and only two clusters observed.

Fig. 8 Posterior estimates of the covariate variables of the Bayesian generalized non-linear multivariate multilevel model, built using annual data. The columns represent soil depths of 0 to 7 cm, 7

to 28 cm, and 28 to 100 cm. Red lines indicate linear regression of precipitation and soil moisture across all ecoregions, with cluster groups represented by three circles. The data point of each ecoregion belongs to a single and non-overlapping cluster.
























Interannual negative dependence was primarily observed in the montane grasslands and shrublands region, where GPP:LST drove this pattern across all three soil layers. All other variables lead to positive dependence (Fig. 8). The long-term trend in the annual-scale Bayesian model revealed strong patterns, with the most significant difference compared to the monthly scale being the influence of Ta:GPP:LST and T<sub>a</sub>:LST, where different ecological zones exhibited substantial variation. Among the multiple variables, T<sub>a</sub> drove the most negative dependence, with the greatest differences observed between ecological zones. In the surface layer, LST alone drove the negative dependence in the mangrove, rock, and ice regions. T<sub>a</sub> drove the negative dependence in tropical and subtropical coniferous forests, lakes, and rock and ice regions. In the middle soil layers, the negative dependence driven by Ta was in temperate forests, arid shrublands, and flooded grasslands and savannas, while it driven by Ta:GPP was in tropical and subtropical moist broadleaf forests. The negative dependence driven by T<sub>a</sub>:LST was fully distributed in quadrant IV. This pattern was observed in regions such as the montane grasslands and shrublands, tropical and subtropical coniferous forests, tropical and subtropical grasslands, savannas, and shrublands; and rock and ice regions. The strongest drivers of negative dependence in the deep layers were GPP:LST and T<sub>a</sub>. The negative dependence driven by GPP:LST was found in the rock and ice regions, Mediterranean forests, woodlands, and scrub, as well as tundra and temperate coniferous forests in quadrant II. The negative dependence driven by Ta was observed in rock and ice regions, lakes, and temperate coniferous forests in quadrant II, and flooded grasslands and savannas in quadrant IV.

# 4. Discussion





























# 4.1 Characteristics of negative dependence areas

In this study, joint distributions of precipitation and soil moisture were constructed using Kendall's  $\tau$  to characterize the nonlinear relationship. Consistent with previous findings, we observed a negative dependence between precipitation and soil moisture, particularly in arid and semi-arid regions (Qing et al., 2023; Yang et al., 2018). At the monthly scale,  $\tau$  values in surface layer were stronger, indicating that seasonal dynamics—such as intermittent rainfall events followed by rapid soil moisture loss through evapotranspiration—likely drive the observed negative correlation. While negative dependence generally decreases with depth, the middle layer shows an unexpectedly low percentage. This layer often corresponds to the main root zone, where stable plant water uptake reduces soil moisture variability and weakens the feedback signal, leading to a few grid cells with significant negative dependence (Thompson et al., 2010). In contrast, the deep soil layers may retain some long-term memory of moisture deficits, especially under prolonged dry conditions, which could contribute to stronger negative dependence than in the more buffered middle layer. On the annual scale, the negative dependence may instead reflect long-term climate feedbacks. In high-latitude regions, for example, Arctic amplification and permafrost thawing can decouple precipitation inputs from effective soil moisture retention, leading to persistent moisture deficits despite increasing precipitation trends. Regions showing negative dependence between precipitation and soil moisture are primarily distributed in arid, semi-arid and cold high-latitude climates. Representative ecosystems include deserts and xeric shrublands, montane grasslands and shrublands, and Arctic tundra. Despite their climatic differences, these ecosystems share key ecohydrological traits, including limited precipitation input, strong evapotranspiration demand, sparse vegetation cover, and low soil moisture retention capacity.

Different from monthly scale, the negative dependence at annual scale is primarily generated in regions such as deserts, xeric shrublands, montane grasslands and

shrublands. These ecosystems are specifically characterized by arid conditions, and particularly sensitive to environmental changes, making them much responsive to longterm climatic variability. In deserts and xeric shrublands, annual precipitation typically falls below 250 mm, while evaporation consistently exceeds rainfall (Lockwood et al., 2006). Vegetation in these regions is dominated by shallow-rooted shrubs, which offer minimal resistance to post-rainfall moisture loss. As a result, soil moisture often declines rapidly following precipitation events, leading to a counterintuitive negative relationship between rainfall and moisture storage. Montane grasslands and shrublands, despite occurring in more topographically complex terrains, also experience dry climatic conditions characterized by low precipitation, high temperatures, and elevated VPD (Olson and Dinerstein, 1998). These factors enhance evapotranspiration, limiting the effectiveness of rainfall in replenishing soil moisture. Consequently, increases in precipitation may coincide with soil moisture decline due to enhanced moisture loss. In contrast, Arctic tundra ecosystems—such as those found in northern North America and Eurasia—are defined by cold temperatures, continuous permafrost, and moderate but ineffective precipitation. Frozen soils impede infiltration, causing much of the precipitation to be lost as surface runoff rather than retained in the soil profile. Dominant vegetation includes mosses, sedges, and dwarf shrubs with shallow root systems, further limiting water uptake and storage (Olson and Dinerstein, 1998; Xue et al., 2021).





























- 4.2 Mechanism of negative dependence between precipitation and soil moisture
- 4.2.1 Energy-Driven Mechanism: LST and Ta-Driven ET Dominance

Negative dependence between precipitation and soil moisture was observed across several dry and cold ecoregions, including deserts and xeric shrublands, montane grasslands and shrublands, tundra. These regions are generally characterized by low precipitation and GPP, limiting vegetation's ability to retain or utilize moisture effectively (Olson and Dinerstein, 1998; Xue and Wu, 2023). In arid ecosystems,

shallow-rooted vegetation and high temperatures result in rapid soil moisture loss following rainfall. In montane environments, stronger warming trends (Pepin et al., 2022) and shallow-rooted vegetation (Stocker et al., 2023) further limit precipitation use, despite increased GPP under warming. Besides, the surface soil induced upward movement of soil water from the middle layer due to the osmotic and matric potential, further contributing to moisture depletion. In semi-arid grasslands, the interaction between soil texture and precipitation patterns further reinforces negative dependence. Brief rainfall events primarily moisten upper clay layers where grass roots concentrate (Sala and Lauenroth, 1985), while well-developed clay horizons restrict deep water percolation and shrub root expansion (Buxbaum and Vanderbilt, 2007). This physical confinement exacerbates water loss when increased GPP and LST enhance evapotranspiration from the shallow moistened zone, intensifying the precipitation-soil moisture decoupling. High temperatures can lead to surface soil sealing, preventing rainfall from effectively entering the root zone. Model simulations confirm that in flat arid regions (Koukoula et al., 2021), such soil barriers promote the "dry soil advantage"—where precipitation triggers runoff rather than infiltration.

The boreal forest and tundra ecosystems, often with permafrost, are temperature-limited systems. Precipitation often falls as snow, which accumulates on the surface. Then, a low LST can cause soil freezing, and the presence of surface withered litter may further insulate the soil, preventing timely moisture replenishment. Permafrost in these regions can lead to surface runoff of some precipitation, preventing effective infiltration into the soil. The geological conditions, such as Karst landforms can also influence the relationship between precipitation and soil moisture.

#### 4.2.2 Biotic-Driven Mechanism: Vegetation Water Use and GPP Dominance

High-altitude ecosystems, especially in the Arctic and Qinghai–Tibetan Plateau, are increasingly affected by warming and variable precipitation (Lamprecht et al., 2018). These changes lead to reduced species abundance and increased GPP (Berauer et al., 2019). In montane grasslands and shrublands, species abundance negatively correlates with soil nutrients and microbial functions (Graham Emily et al., 2024). Rising LST

and extreme precipitation reduce microbial biomass and release soil minerals (Siebielec et al., 2020), intensifying light competition and lowering ecosystem stability. Biodiversity loss decreases soil water capacity, with some of these regions at high risk of water erosion (Straffelini et al., 2024).

Soil moisture reduction in the surface and middle layer is mainly driven by root water uptake under high LST and GPP. Roots shift absorption to deeper layers during droughts (Yadav Brijesh et al., 2009). In dry seasons, plants in grasslands and shrublands retain leaves to support evaporative cooling (Prior et al., 1997), this strategy also seen in deserts and xeric shrublands, where winter precipitation and freezing reduce surface moisture. Even during rainfall, soil moisture may decline due to evapotranspiration, runoff, and plant uptake (Tomlinson et al., 2013), creating a negative precipitation—soil moisture relationship. Canopy interception also limits infiltration (Zhong et al., 2022). However, in high-latitude ecosystems like boreal forests and tundra, warming mitigates cold limitations, allowing precipitation to increase soil moisture, shifting the relationship to positive.

Negative dependence in mid-to-deep soil layers can occur when a single factor dominates, limiting ecosystem compensation (Jarvis, 2011; Taylor and Klepper, 1979). In contrast, positive dependence may arise from synergistic interactions between GPP and LST. Higher GPP can reflect deeper root systems or improved water-use efficiency, while increased LST may enhance soil moisture release and promote water availability together (Wang et al., 2008). This interaction may strengthen ecosystem feedbacks—e.g., higher GPP can improve soil structure through biomass and organic matter, boosting water retention (Chen et al., 2025). Such synergy can offset LST-driven evapotranspiration and enhance ecosystem resilience, particularly through freeze—thaw processes in cold regions.

## 4.3 Data reliability

In this study, multiple observational datasets were employed to reduce modeldriven uncertainty and enhance data reliability. CRU TS, ESA CCI, and GPCP were selected due to their direct reliance on ground-based or satellite observations, in contrast to the model-based ERA5-Land product. Although ERA5 does offer a wide range of meteorological variables, it can introduce model uncertainties. Therefore, the datasets used in this study have independent source, which can avoid the potential false relationships between soil moisture and precipitation that may be caused by the same model architecture and input parameters. To investigate spatial heterogeneity, all data were spatially aggregated by ecoregion boundaries from the Conservation Biology Institute. These boundaries may introduce regional biases, which should be considered when interpreting the results.

The copula method can access the dependence between different time series, after removing influences of the conditional means and variances as well as marginal distributions (Durante et al., 2025; Neumeyer et al., 2019). In this study, although precipitation—soil moisture dependence was assessed across different time scales, the monthly series were not de-seasonalized. As a result, the residual seasonal signals may influence short-term dependence structures. This limitation will be addressed in future work through seasonal adjustment. In the Bayesian modeling, GPP, LST, and air temperature were examined as drivers of negative dependence. Evapotranspiration was excluded due to its dependence on both soil moisture and temperature. We acknowledge that additional factors—such as wind, topography, and soil physical properties—may also modulate precipitation—soil moisture coupling but were not in the scope of this analysis. Future research incorporating these variables would provide a more comprehensive understanding of the underlying mechanisms.

## 5. Conclusion

This study explored the dependence relationships between precipitation and soil moisture at depths of 0 to 7 cm, 7 to 28 cm, and 28 to 100 cm from 2000 to 2019, by examining the control effect of precipitation volume, precipitation frequency, and evapotranspiration on soil moisture. Bayesian models were used to analyze the driving factors and relative contribution in the dependence of soil moisture to precipitation in different time scales and ecoregions of the Northern Hemisphere. The results showed that, the negative dependence proportion reached 19.2%, 0.7%, and 2.3% at monthly

scale, while it was 3.0%, 4.0%, and 8.6% at annual scale, respectively, for the three soil layers. Our studies have new insight for the dependence of soil moisture to precipitation varying in different ecoregions. We concluded that, precipitation volume predominantly controlled soil moisture in the Boreal forest/taiga, temperate grasslands, savannas, and shrublands, while precipitation frequency primarily controlled soil moisture in the high-latitude regions of the Northern Hemisphere. The combined influence of evapotranspiration and precipitation exhibited clear seasonal patterns. While evapotranspiration is known to dominate soil moisture dynamics during the growing season (Kozii et al., 2020), this study quantified that this dominance are with regression coefficients more than 75% of the total sum of the three covariates. In contrast, precipitation volume played a more significant role in the surface and middle layer of non-growing season, with areas under strong univariate control accounting for over 40% of the total area. Additionally, the influence of precipitation frequency on soil moisture increased with latitude, the proportion of the regression coefficient averaging from 36.5% to 91.3%, highlighting a shift in controlling factors across climatic gradients.

For the factor driving the dependence of soil moisture to precipitation, this study found that the negative dependences were distributed across temperate grasslands, savannas, shrublands, deserts, xeric shrublands, and tundra, primarily driven by LST and Ta:GPP interactions. These negative dependences were mainly attributed to the seasonality of precipitation in arid and semi-arid areas and the freeze—thaw processes in the soil, which hinder effective moisture replenishment, especially during winter when soil freezing prevents rainwater infiltration. In the intermediate and deep soil layers, negative dependences were primarily driven by single variables, whereas positive dependences resulted from multivariate interactions, likely due to the lack of compensatory mechanisms when a single variable dominated, or the enhancement of ecosystem feedbacks when both GPP and LST interacted. Additionally, when the ecosystem is simultaneously driven by GPP and LST, greater resilience may be exhibited.

At the annual scale, the negative dependences were mainly in the montane

grasslands and shrublands region (Wei et al., 2008). This study further revealed that this 662 negative dependence increased with soil depth, and were driven by the GPP:LST 663 interaction across all three soil layers. A possible explanation is the long-term 664 variability in precipitation and temperature, which may have influenced 665 geomorphology, vegetation structure, and soil water retention capacity. 666 667 **Data availability** 668 The ERA5-Land soil moisture dataset (https://doi.org/10.24381/cds.e2161bac) was 669 obtained from the Copernicus Climate Data Store (accessed on 18 March 2024). The 670 GPCP precipitation dataset (https://doi:10.7289/V5RX998Z.) was obtained from the 671 NOAA National Centers for Environmental Information (accessed on 11 March 672 2024). The Gross primary production dataset (https://doi.org/10.5194/essd-14-1063-673 2022) was obtained from TU Wien Research Data Repository (accessed on 23 674 October 2023). The CRU TS v4.07 air temperature dataset 675 (https://doi.org/10.1038/s41597-020-0453-3) was obtained from the Climatic 676 Research Unit (accessed on 20 August 2023). The ESA CCI Land Surface 677 678 Temperature dataset (https://dx.doi.org/10.5285/a7e811fe11d34df5abac6f18c920bbeb) was obtained from 679 the Centre for Environmental Data Analysis (accessed on 27 August 2024). GLEAM 680 Evapotranspiration data (https://doi.org/10.5194/gmd-10-1903-2017) was obtained 681 from the GLEAM project (accessed on 19 March 2024). Terrestrial Ecoregions 682 dataset (https://doi.org/10.1641/0006-3568(2001)051[0933:TEOTWA]2.0.CO;2) was 683 obtained from the World Wildlife Fund (accessed on 5 September 2024). 684 685 **Author Contributions** 686 SX: Conceptualization, Data curation, Formal analysis, Methodology, Software, 687 Validation, Visualization, Writing - Original Draft, Writing - Review & Editing. GW: 688

Conceptualization, Funding acquisition, Investigation, Supervision, Writing - Original

Draft, Writing - Review & Editing.

| 691 |                                                             |
|-----|-------------------------------------------------------------|
| 692 | <b>Conflicts of Interest</b>                                |
| 693 | The authors declare that they have no conflict of interest. |
| 694 | Financial Support                                           |

This study was funded by the National Key Research and Development Program of China (2022YFF0801302), the National Natural Science Foundation of China (41930970 and 42077421).

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
