# Peer review of "A Study of the Dependence between Soil Moisture and"

_EGUsphere, 2025_

## Author Comment (AC2)

**Journal:** Hydrology and Earth System Sciences

**Title:** A Study of the Dependence between Soil Moisture and Precipitation in different Ecoregions of the Northern Hemisphere

**Author(s):** Shouye Xue; Guocan Wu

**Manuscript No.:** EGUSPHERE-2025-762

We highly appreciate the anonymous reviewer for the very helpful and insightful comments that lead to the significant improvement of the quality of this manuscript. We have checked our work carefully according to these comments and made the requested changes. For point to point response, we indicate the comments and use blue font for our responses.

**Anonymous Referee #2**

The authors examine the dependences between soil moisture and precipitation, and their drivers across the northern hemisphere. They find substantial negative dependences, which are mostly attributed to evapotranspiration and vegetation conditions. The topic is intriguing and the methodology interesting. However, I think this paper could benefit from refinement in areas like novelty declaration, method justification, presentation quality, and enhanced supporting evidence for conclusions. Therefore, I would recommend a major revision.

Response: We sincerely thank you for your valuable comments. Based on your suggestions, the Results and Discussion sections were reorganized for improving clarity. Some quantitative summery was added to the Abstract and Conclusion sections. Figure 2 was re-plotted and Figures S3 and S4 were added for considering the time lag. These modifications have improved the overall quality of the manuscript.

1. Below are major concerns that expect to authors to address in the revised manuscript. The authors need to identify the research gaps and specify any novel findings or methodology not reported in earlier studies. Negative correlations between soil moisture and precipitation and their causes have previously been identified, a fact acknowledged by the authors (Line 66-73). The authors seem to claim their novelty in terms of climate change and climate extremes (Line 101-106). However, I find the two points only loosely related to this study.

Response: Thanks for your comment. We have rewritten the related sentences to identify the research gaps and specify novel in abstract as follows.

"Although previous studies have identified the mechanisms of soil moisture variation across different time scales (Shen et al. 2018; Vidana Gamage et al. 2020), the interaction among precipitation, evapotranspiration and soil water under climate change may have changed. The dependence of soil moisture to precipitation and its interactions with evapotranspiration under conditions of climate change require further investigation."

"This study clarifies the regional differences and driving mechanisms of the

negative correlation between precipitation and soil moisture."

Shen, S., and Coauthors, 2018: Persistence and Corresponding Time Scales of Soil Moisture Dynamics During Summer in the Babao River Basin, Northwest China. Journal of Geophysical Research: Atmospheres, 123, 8936-8948.

Vidana Gamage, D. N., A. Biswas, and I. B. Strachan, 2020: Scale and location dependent time stability of soil water storage in a maize cropped field. CATENA, 188, 104420.

2. There is a lack of quantitative summary of the findings throughout the paper, especially in the abstract and conclusion sections. For instance, the authors should indicate the proportion of positive/negative correlations across various soil layers/ecoregions and quantify the contribution of controlling factors.

Response: Thanks for your suggestion. We have revised abstract and conclusion, and the quantitative summary was added.

"……Nonlinear negative dependencies of soil moisture to precipitation were revealed. The monthly scale negative dependence proportion reached 19.2%, 0.7%, and 2.3%, while the annual scale was 1.8%, 3.8% and 6.4%, respectively."

"……Among them, temperature most strongly drives the deep-layer negative dependence in the tundra over annual scale, with the absolute difference between the posterior estimates of temperature on precipitation and soil moisture reaching 0.32."

"……The results suggest that, at the monthly scale, negative dependence proportion reached 19.2%, 0.7%, and 2.3%, while the annual scale was 1.8%, 3.8% and 6.4%, respectively."

3. The Results and Discussion section reads too imbalance. Currently, there is a lack of reasoning of the findings shown in the Results section, making the results a bit dull to read. The reasoning in Discussion is too spread and redundant, causing readers having to flip between the two sections. Also, I think a schematic diagram might help.

Response: We appreciate your comment about the readability of the manuscript. We have substantially revised the structure of the Results and Discussion sections to enhance logical flow and readability. In particular, we have reorganized subsection 4.3

based on the driving mechanisms and strengthened the interpretation of our findings directly within the Results section to reduce redundancy. We hope these changes can address your concerns effectively.

4.  The ridge regression and Section 3.2 seem off topic, as the main scope is to study the dependences between precipitation and soil moisture as well as their drivers. As a key driver of the dependences, why ET is not added to the Bayesian model. The soil property, another key controlling factor according to the authors, is also not considered in the Bayesian model as well (Line 116).

Response: The ridge regression model was established to quantify the driving intensity of precipitation-evapotranspiration on soil water, which is a complementary analysis for joint distribution. Considering evapotranspiration is generally correlated with soil moisture and air temperature, while soil moisture is used as the dependent variable in the Bayesian model, air temperature and ground temperature are considered as driving factors. So it is not necessary to consider ET as the driving factor separately. Soil factors include many factors such as soil depth, soil texture, etc., and this study mainly explores the dependence in different depths. These were further explained in Section 4.3 in the revised version.

"In addition to the factors discussed in this study, other variables such as wind patterns and topography may also influence the negative dependence between precipitation and soil moisture. Soil properties—such as texture, organic matter content, and hydraulic conductivity—represent another set of important controls that were not explicitly included in the current Bayesian models. While this study provides a foundational analysis of the negative dependencies across different ecoregions, future research should explore these additional environmental factors to gain a more comprehensive understanding of the mechanisms underlying precipitation–soil moisture interactions."

5.  The authors should justify their use of eco-region boundaries over the more well-known climate region, e.g., Köppen climate classification system.

Response: Ecoregions are divided based on an integrated consideration of vegetation types, soils, substrate, and climate. Compared to climate zones, they can better capture the heterogeneity of regional water feedback processes. Therefore, ecoregion boundaries were used instead of climate zones. The explanation for this choice was added in the revised manuscript.

"In this study, the ecoregion boundaries rather than Köppen climate zones were used to investigate the spatial patterns of precipitation–soil moisture feedbacks. Ecoregions are divided based on a combination of factors including vegetation types, soil characteristics, substrate, and climate conditions. This multi-factor approach allows ecoregions to better reflect ecological and hydrological processes than classifications based solely on climate variables. Since soil moisture dynamics and their feedbacks with precipitation are strongly influenced by vegetation structure, root systems, and edaphic properties, the ecoregions can provide a more mechanistic and spatially relevant framework for our analysis."

6. The dependence between soil moisture and precipitation might not be concurrent, and could have a lag time. There are little consideration and discussion of this point.

Response: We appreciate your insightful comment regarding the potential non-concurrent relationship between precipitation and soil moisture, and we fully agree with this point.

In the revised version, we have re-evaluated the dependence between precipitation and soil moisture by incorporating time-lagged effects. Specifically, for each grid cell, a maximum lag of up to 12 months was used to calculate the lagged correlation between precipitation and soil moisture. Then the optimal lag for each grid cell was determined by identifying the time lag that yielded the maximum Kendall's tau within this 0–12 month window. To assess model adequacy, the Akaike Information Criterion (AIC) was calculated for each lag.

[Figure]

Fig. S3 The AIC value for each grid in the selection of copula function.

[Figure]

Fig. S4 The estimated number of lagged month for each grid in the the Kendall's tau correlation.

The related text was added in revise manuscripts as follows.

"To address the potential delayed response of soil moisture to precipitation, lagged correlation analysis was conducted. For each grid cell, the AIC value was calculated to select copula function (Fig. S3), then the Kendall's tau correlation was

calculated between precipitation and soil moisture with time lags ranging from 0 to 12 months (Fig. S4). The lag corresponding to the maximum absolute correlation was identified as the optimal lag."

[Figure]

Fig. 2 Spatial distribution of Kendall's tau ($\tau$), the upper tail dependence ($\lambda_U$), and the lower tail dependence ($\lambda_L$) on the 0.25 ° × 0.25 °grids between monthly precipitation volume and soil moisture with the time lag during 2000 to 2019. The three columns are for the soil moisture from depths of 0 to 7 cm, 7 to 28 cm, and 28 to 100 cm, respectively.

"The copula analysis of monthly average soil moisture and total monthly precipitation volume revealed a clear negative dependence at all three soil depths (a2, b2, and c2; Fig. 2). The percentages of grid cells exhibiting negative dependence at these depths were 19.2%, 0.7%, and 2.3%, respectively. The negative dependence between precipitation and soil moisture is more prevalent in the surface soil layer, where the grid cells exhibiting are more widespread. In contrast, at the middle and deep soil layers, these negative dependence patterns are primarily confined to the margins of the Sahara desert, the montane grasslands and shrublands, and parts of the deserts and xeric shrublands regions. In the surface layer, the negatively dependent grid patches are more spatially scattered, mainly distributed across the tundra, montane grasslands and shrublands, deserts and xeric shrublands, as well as the tropical and subtropical moist broadleaf forests.

Regions exhibiting high λL values were primarily located in the deserts and xeric Shrublands, as well as in parts of India, where λL reached values as high as 0.99(a1, b1, c1; Fig. 2). With increasing soil depth, λL values gradually increased across the

Eurasian continent. Similarly, $\lambda_U$ exhibited a clear reduction in spatial extent with increasing soil depth, with the majority of these regions located in the temperate broadleaf and mixed forests and the southern margin of the Sahara desert. With increasing soil depth, $\lambda U$ values consistently decreased, resulting in a lack of clear correspondence between these regions and specific ecological zones (a3, b3, c3; Fig. 2)."

7.  I also have concerns about the time scale. I agree with reviewer #1 that the time scale (monthly, seasonal, annual) should be unified. Since the authors did not eliminate seasonal variations from monthly data, seasonal signals affect the monthly-scale results. The patterns and mechanisms during seasons appear clearer. I would suggest the authors to narrow the analyses by only focusing on one or two scales.

Response: Thanks for your suggestion. The results of annual scale have been added in Section 3.2 as follows.

"At the annual scale, results are consistent with those at the monthly scale, with precipitation amount continuing to exert a dominant influence across all three soil depth layers, accounting for more than 40% of the total area (Fig. 6). The spatial extent of areas dominated by precipitation amount, precipitation frequency, and evapotranspiration remains largely consistent with that observed at the monthly scale. In terms of spatial distribution, regions dominated by precipitation frequency are still primarily located in high-latitude areas, particularly in Greenland and the Queen Elizabeth Islands, although no distinct ecological zone patterns are observed in these areas. Regions dominated by precipitation amount are mainly distributed across boreal forests, temperate grasslands, savannas and shrublands, temperate broadleaf and mixed forests, as well as tropical and subtropical moist broadleaf forests. In temperate regions, soil moisture is primarily controlled by precipitation amount due to moderate temperatures and limited rainfall, making substantial precipitation inputs essential for soil moisture replenishment. In contrast, tropical and subtropical regions experience high temperatures and intense evapotranspiration, requiring substantial precipitation to maintain a water balance."

[Figure]

Fig. 6 Ternary map of factors controlling soil moisture at annual scale, for the period 2000 to 2019. The bottom-left histogram in the subgraph represents the proportion of grid cells where one variable exerts strong univariate control (with a regression coefficient greater than 75% of the total sum of the three variables), suggesting that soil moisture was predominantly controlled by that specific variable."

Minor comments:

1) Line 489: the "Arctic amplification" appears abruptly. How are the climate pattern associated with the dependences? Why not other climate patterns?

Response: Our intention means that, the heterogeneity in global warming among different regions can affects GPP and drives precipitation-soil water feedback. However, climate pattern are not the focus of this study, which aims to explore the driving characteristics of GPP, air temperature, and ground temperature in different regions.

2) Line 517-519: not clear, need rephrase.

Response: The sentence was deleted in revised manuscript.

3) Line 747-748: need quantitative measures to support this point.

Response: Revised.

"Evapotranspiration was the dominant driver of soil moisture dynamics during the growing season, with a regression coefficient proportion greater than 75%. In contrast, precipitation volume played a more significant role in the surface and middle layer of non-growing season, with areas under strong univariate control accounting for over 40% of the total area. Additionally, the influence of precipitation frequency

on soil moisture increased with latitude, the proportion of the regression coefficient averaging from 36.5% to 91.3%, highlighting a shift in controlling factors across climatic gradients."

4) Line 764-769: these reasoning needs quantitative support.

Response: Thanks for your suggestion. Since this point is not the focus of our study, we have revised this sentence in the manuscript to better reflect our reasonable speculation.

"A possible explanation is the long-term variability in precipitation and temperature, which may have influenced geomorphology, vegetation structure, and soil water retention capacity."

---

## Author Response (AR1)

Journal: Hydrology and Earth System Sciences

Title: A Study of the Dependence between Soil Moisture and Precipitation in different

Ecoregions of the Northern Hemisphere

Authors: Shouye Xue; Guocan Wu

Manuscript No.: EGUSPHERE-2025-762

We highly appreciate the editor and anonymous reviewers for the very helpful and insightful comments that lead to the significant improvement this manuscript. We have checked our work carefully according to these comments and made the requested changes. In the revised version, the Results and Discussion sections were reorganized for improving clarity. Some quantitative summery was added to the Abstract and Conclusion sections. We also added Figure 6 for reflecting the results on annual scale and Figures S1 and S2 in revised manuscript for reflecting the accuracy of the copula functions.

Below we indicate the comments and use blue font for our responses. The corresponding revised texts are also used blue font in the revised version of our manuscript.

**Editior**

Notification to the authors:

1) In the "Data availability" section, please consider indicating the doi numbers

(where available) instead of weblinks.

Response: Revised.

2) For the section "Author contribution", please use initials only for the authors

instead of full names.

Response: Revised.

3) Most figures (also supplement figures) are very pixelated. Please provide figure

images with a better quality.

Response: Thanks, the pixelated figures have been replaced.

4) You uploaded a supplement file but do not mention the supplement in the

manuscript text. Please consider mentioning the supplement file as reads might

otherwise not be aware of the supplementary material.

Response: Thanks, some sentences have been added to reference the supplementary

material.

**Anonymous Referee #1**

The study investigates the relationship between soil moisture and precipitation, but a lot needs to be clarified. First, it is stated in the abstract that soil moisture is jointly affected by precipitation and evapotranspiration, but there is no description of evapotranspiration in the abstract. Secondly, the respective roles of Ridge regression models and Bayesian generalized non-linear multivariate multilevel models in attribution need to be explained. What causes the differences in dependencies between land cover types? How do these differences come about? There is no consensus on what common features these land covers have. Finally, the effects on seasonal scales and interannual scales look more like the usual conclusions, and it is not clear that this work finds something new based on these traditional results.

Response: Thanks for your thorough review, and we appreciate for your insightful comments. In the response, we have highlighted the major findings of the study, reorganized the logical flow among the three key components, and revised both the Results and Discussion sections. Additionally, we have added Figure 6 for reflecting the results on annual scale and Figures S1 and S2 in revised manuscript for reflecting the accuracy of the copula functions. We hope that the following point to point response can address your concerns.

I suggest a major revision. Please see my comments below:

**Major Comments**

1. In the introduction, the linear or nonlinear relationship here is a model for estimating soil moisture by precipitation, whereas copula is a distribution function, they should not be compared together. Ridge regression is an important method in the abstract, but it is not mentioned in the introduction. What role does ridge regression play?

Response: This study employs the joint distribution of precipitation and soil moisture to capture their nonlinear relationship. The copula function is a multivariate statistical method that can describe the dependency relationships between multiple variables

through their joint distribution as a compound event. The statistic Kendall's  $\tau$  generated form the copula function can be served as an effective measurement, if the relationship between precipitation and soil moisture is nonlinear. Therefore, copula function approach is used to investigate the nonlinear dependence between precipitation and soil moisture in this study. We have added further clarification on this point in the manuscript.

Ridge regression is used in this study to quantify the relative influence of precipitation amount, precipitation frequency, and evapotranspiration on soil moisture. We have included a corresponding explanation in the Introduction.

"Accordingly, the ridge regression models for precipitation amount, precipitation frequency, evapotranspiration, and soil moisture can be used to quantify the relative influence of precipitation and evapotranspiration on soil moisture. As an improvement of the least squares estimation method, it can handle the multi-collinearity problems of the covariates, although it is usually biased." (Lines 109-114)

2. In the material and method, the joint probability of copula considers soil moisture and precipitation, ridge regression considers precipitation and evapotranspiration to predict soil moisture, Bayesian generalized non-linear multivariate multilevel models consider GPP, LST, and temperature to predict soil moisture and precipitation, what is the relationship between these three methods that seem to be simply spliced together. Why choose these models and how accurate are they in the simulation?

Response: Previous studies have found a negative correlation between precipitation and soil moisture; however, such findings often lack spatial generality. To address this, the first part of our study establishes a joint distribution to capture the nonlinear monotonic relationship (dependence) between precipitation and soil moisture, confirming the consistent presence of this negative dependence across multiple temporal scales. The second part investigates how changes in precipitation characteristics influence the control exerted by precipitation and evapotranspiration on soil moisture. A ridge regression model is constructed to quantify whether the observed negative dependence between precipitation and soil moisture across different regions is primarily driven by

precipitation or by evapotranspiration. This model has a particular focus on the strength of evapotranspiration, which is treated as a driving factor. The third part explores the roles of air temperature, land surface temperature, and GPP in modulating the dependence between precipitation and soil moisture, and identifies region-specific patterns. The Bayesian nonlinear multivariate multilevel model is particularly employed in this study, since it can accurately capture both individual and interactive effects of multiple drivers on the regulation of precipitation—soil moisture relationships.

To ensure model accuracy, the MCMC samples were extracted from the Bayesian model and the Rhat values was computed for convergence diagnostics. Furthermore, to ensure the statistical soundness of the selected copula function, we calculated the Akaike Information Criterion (AIC) and time lag for each grid and use it to verify the appropriateness of the chosen copula function and time lag.

The relevant information on model accuracy has been included in the Appendix as follows. The related sentences were added to the manuscript.

"To address the potential delayed response of soil moisture to precipitation, lagged correlation analysis was conducted. For each grid cell, the AIC value was calculated to select copula function (Fig. S1), as shown in the supplementary file. Then the Kendall's tau correlation was calculated between precipitation and soil moisture with time lags ranging from 0 to 12 months (Fig. S2). The lag corresponding to the maximum absolute correlation was identified as the optimal lag." (Lines 259-265)

Fig. S1 The AIC value for each grid in the selection of copula function.

Fig. S2 The estimated number of lagged month for each grid in the Kendall's tau correlation.

3. In Section 3.1, for example in northwest Africa, why is there a negative dependence between the soil moisture at the first layer and precipitation while a positive dependence between the soil moisture at the second layer and precipitation, and what causes the difference between the different layers? Is there any connection

between the result expressed by  $\lambda U/\lambda L$  and Kendall's tau, and why do many grids have no value in the result expressed by  $\lambda U/\lambda L$ ?

Response: In this study, different copula methods were applied to construct the joint distribution across different grid cells. However, we selected the method with high goodness-of-fit, even though some of these copula functions do not support the estimation of  $\lambda U$  and  $\lambda L$ . Therefore, the Kendall's  $\tau$  as the primary indicator was emphasized and the calculation of  $\lambda U$  and  $\lambda L$  could be omitted in regions where the applied method does not support their estimation.

The results in Sections 3.2 and 3.3 further indicated that the variation in correlation across different soil depths is driven by multiple factors, including air temperature, vegetation root distribution, and ecosystem characteristics. In the joint distribution framework, Kendall's  $\tau$  characterizes the overall monotonic relationship of the full time series, while  $\lambda U$  and  $\lambda L$  represent tail dependence under extreme conditions, capturing the dependence between precipitation and soil moisture during extreme drought or extreme wetness.

4. In Section 3.1, the monthly scale and annual scale are used, but in Section 3.2, the monthly scale and seasonal scale are used, so it is recommended to unify the comparison scale.

Response: Thanks for your suggestion. The results of annual scale have been added in Section 3.2 as follows. (lines 403-417)

"At the annual scale, precipitation amount exerts a dominant influence across all three soil depth layers, accounting for more than 40% of the total area (Fig. 6). The spatial extent of areas dominated by precipitation amount, precipitation frequency, and evapotranspiration remains largely consistent with that observed at the monthly scale. The regions dominated by precipitation frequency are still primarily located in high-latitude areas, particularly in Greenland and the northern parts of Canada, although no distinct ecological zone patterns are observed in these areas. Regions dominated by precipitation amount are mainly distributed across boreal forests, temperate grasslands, savannas and shrublands, temperate broadleaf and mixed forests,

as well as tropical and subtropical moist broadleaf forests. In temperate regions, soil moisture is primarily controlled by precipitation amount due to moderate temperatures and limited rainfall, making substantial precipitation is essential for soil moisture replenishment. In contrast, tropical and subtropical regions experience high temperatures and intense evapotranspiration, requiring substantial precipitation to maintain a water balance."

Fig. 6 Ternary map of factors controlling soil moisture at annual scale, for the period 2000 to 2019. The bottom-left histogram in the subgraph represents the proportion of grid cells where one variable exerts strong univariate control (with a regression coefficient greater than 75% of the total sum of the three variables), suggesting that soil moisture was predominantly controlled by that specific variable.

5. There are too many descriptions in section 3.3, and scatters of different land cover types in figures 6 and 7 are not clear. The large number of listed results makes it difficult to distinguish the commonalities and differences between different land cover types, and why there are differences between different soil layers. Part of the discussion should be summarized in the results, and the discussion should add references.

Response: Thanks for your comment. Section 3.3 has been re-written and Figures 6 and 7 has been re-plotted as follows, which were numbered as Figures 7 and 8 in the revised version. (Lines 426-481)

**"3.3 Drivers of negative dependencies between soil moisture and precipitation**

[revised manuscript text omitted]

Some of the discussion has be moved the results, and the references have been added to the discussion. Please see our responses to the following comments.

6. The first paragraph in Section 4.1 repeats the results, which should add references to compare and explain why this is the case. The second paragraph of the discussion is more like an introduction to land cover types but does not explain why.

Response: Thanks for your suggestion. Section 4.1 has revised as follows. (Lines 484-519)

**"4.1 Characteristics of negative dependence areas**

In this study, joint distributions of precipitation and soil moisture were constructed using Kendall's  $\tau$  to characterize the nonlinear relationship. Consistent

with previous findings, we observed a negative dependence between precipitation and soil moisture, particularly in arid and semi-arid regions (Qing et al., 2023; Yang et al., 2018). At the monthly scale,  $\tau$  values in surface layer were stronger, indicating that seasonal dynamics—such as intermittent rainfall events followed by rapid soil moisture loss through evapotranspiration—likely drive the observed negative correlation. On the annual scale, the negative dependence may instead reflect long-term climate feedbacks. In high-latitude regions, for example, Arctic amplification and permafrost thawing can decouple precipitation inputs from effective soil moisture retention, leading to persistent moisture deficits despite increasing precipitation trends. Regions showing negative dependence between precipitation and soil moisture are primarily distributed in arid, semi-arid and cold high-latitude climates. Representative ecosystems include deserts and xeric shrublands, montane grasslands and shrublands, and Arctic tundra. Despite their climatic differences, these ecosystems share key ecohydrological traits, including limited precipitation input, strong evapotranspiration demand, sparse vegetation cover, and low soil moisture retention capacity.

In deserts and xeric shrublands, annual precipitation typically falls below 250 mm, while evaporation consistently exceeds rainfall (Lockwood et al., 2006). Vegetation in these regions is dominated by shallow-rooted shrubs, which offer minimal resistance to post-rainfall moisture loss. As a result, soil moisture often declines rapidly following precipitation events, leading to a counterintuitive negative relationship between rainfall and moisture storage. Montane grasslands and shrublands, despite occurring in more topographically complex terrains, also experience dry climatic conditions characterized by low precipitation, high temperatures, and elevated VPD (Olson and Dinerstein, 1998). These factors enhance evapotranspiration, limiting the effectiveness of rainfall in replenishing soil moisture. Consequently, increases in precipitation may coincide with soil moisture decline due to enhanced moisture loss. In contrast, Arctic tundra ecosystems—such as those found in northern North America and Eurasia—are defined by cold temperatures, continuous permafrost, and moderate but ineffective precipitation. Frozen soils impede

infiltration, causing much of the precipitation to be lost as surface runoff rather than retained in the soil profile. Dominant vegetation includes mosses, sedges, and dwarf shrubs with shallow root systems, further limiting water uptake and storage (Olson and Dinerstein, 1998; Xue et al., 2021)."

7. The meltwater discussed in 4.2 is even an important part of the abstract, but the meltwater is not used in the results. The discussion should be based on the main content of the results, and the discussion should also consider the geological conditions, such as karst landform, in addition to the influence of vegetation.

Response: We acknowledge the limitations of the ERA5-Land dataset in capturing snow and permafrost dynamics, particularly in high-latitude regions. These limitations could affect the accuracy of snowmelt estimation and its influence on soil moisture (Kouki et al., 2023). This study does not intend to discuss "meltwater", since the main objective is to investigate how changes in LST and Ta influence the phase of precipitation (e.g., rain vs. snow) and how these changes affect water availability.

We also agree with you that geological conditions such as karst topography may influence the spatial patterns of precipitation—soil moisture relationships. We have added some discussion in Section 4.2 in the revised version.

"The geological conditions, such Karst landforms can also influence the relationship between precipitation and soil moisture." (Lines 550-551)

8. It is suggested that Section 4.3 be parted in different sections according to different mechanisms.

Response: Thanks for your suggestion. Section 4.2 and 4.3 were merged together and it has been rewritten and re-numbered as follows. (Lines 523-582)

"4.2.1 Energy-Driven Mechanism: LST and Ta-Driven ET Dominance

[revised manuscript text omitted]

**Minor Comments**

**1. What does dependence mean?**

Response: Nonlinear and asymmetric correlations in joint distributions are generally defined as dependence (Dette et al., 2013), we have explained it in manuscript as

follows.

"This kind of nonlinear and asymmetric correlation is generally referred to as

'dependence'." (Lines 77-78)

2. Line 287: What are the multivariate mixed effects, and why do these variables

combine?

Response: Multivariate mixed effects refer to the interaction effect of multiple

variables. Specifically, the environmental elements in the ecosystem can restrain and

promote each other, and ultimately produce the same or opposite effects as the single

variable drive. Therefore, this study considered the single effect of different driving

factors and the multivariate interaction effect.

Line 519: The results about arid areas should be added after the reference to compare.

Response: The sentence was deleted in revised manuscript.

Line 532: The figures in the results should be marked here.

Response: Revised.

Response: We appreciate your insightful comment regarding the potential non-concurrent relationship between precipitation and soil moisture, and we fully agree with this point.

In the revised version, we have re-evaluated the dependence between precipitation and soil moisture by incorporating time-lagged effects. Specifically, for each grid cell, a maximum lag of up to 12 months was used to calculate the lagged correlation between precipitation and soil moisture. Then the optimal lag for each grid cell was determined by identifying the time lag that yielded the maximum Kendall's tau within this 0–12 month window. To assess model adequacy, the Akaike Information Criterion (AIC) was calculated for each lag.

Fig. S1 The AIC value for each grid in the selection of copula function.

Fig. S2 The estimated number of lagged month for each grid in the Kendall's tau correlation.

The related text was added in revise manuscripts as follows.

"To address the potential delayed response of soil moisture to precipitation, lagged correlation analysis was conducted. For each grid cell, the AIC value was calculated to select copula function (Fig. S1), as shown in the supplementary file.

Then the Kendall's tau correlation was calculated between precipitation and soil moisture with time lags ranging from 0 to 12 months (Fig. S2). The lag corresponding to the maximum absolute correlation was identified as the optimal lag." (Lines 259-265)

Fig. 2 Spatial distribution of Kendall's tau  $(\tau)$ , the upper tail dependence  $(\lambda_U)$ , and the lower tail dependence  $(\lambda_L)$  on the  $0.25~\times 0.25~$  °grids between monthly precipitation volume and soil moisture with the time lag during 2000 to 2019. The three columns are for the soil moisture from depths of 0 to 7 cm, 7 to 28 cm, and 28 to 100 cm, respectively.

"The copula analysis of monthly average soil moisture and total monthly precipitation volume revealed a clear negative dependence at all three soil depths (Fig. 2(a2, b2, c2)). The percentages of grid cells exhibiting negative dependence at these depths were 19.2%, 0.7%, and 2.3%, respectively. The negative dependence between precipitation and soil moisture is more prevalent in the surface soil layer, where the grid cells exhibiting are more widespread. In contrast, at the middle and deep soil layers, these negative dependence patterns are primarily confined to the margins of the Sahara desert, the montane grasslands and shrublands, and parts of the deserts and xeric shrublands regions. In the surface layer, the negatively dependent grid patches are more spatially scattered, mainly distributed across the tundra, montane grasslands and shrublands, deserts and xeric shrublands, as well as the tropical and subtropical moist broadleaf forests.

Regions exhibiting high  $\lambda L$  values were primarily located in the deserts and xeric Shrublands, as well as in parts of India, where  $\lambda L$  reached values as high as 0.99 (Fig.

2(a1, b1, c1)). With increasing soil depth,  $\lambda L$  values gradually increased across the Eurasian continent. Similarly,  $\lambda_U$  exhibited a clear reduction in spatial extent with increasing soil depth, with the majority of these regions located in the temperate broadleaf and mixed forests and the southern margin of the Sahara desert. With increasing soil depth,  $\lambda U$  values consistently decreased, resulting in a lack of clear correspondence between these regions and specific ecological zones (Fig. 2(a3, b3, c3))." (Lines 329-349)

7. I also have concerns about the time scale. I agree with reviewer #1 that the time scale (monthly, seasonal, annual) should be unified. Since the authors did not eliminate seasonal variations from monthly data, seasonal signals affect the monthly-scale results. The patterns and mechanisms during seasons appear clearer. I would suggest the authors to narrow the analyses by only focusing on one or two scales.

Response: Thanks for your suggestion. The results of annual scale have been added in Section 3.2 as follows.

"At the annual scale, results are consistent with those at the monthly scale, with precipitation amount continuing to exert a dominant influence across all three soil depth layers, accounting for more than 40% of the total area (Fig. 6). The spatial extent of areas dominated by precipitation amount, precipitation frequency, and evapotranspiration remains largely consistent with that observed at the monthly scale. In terms of spatial distribution, regions dominated by precipitation frequency are still primarily located in high-latitude areas, particularly in Greenland and the Queen Elizabeth Islands, although no distinct ecological zone patterns are observed in these areas. Regions dominated by precipitation amount are mainly distributed across boreal forests, temperate grasslands, savannas and shrublands, temperate broadleaf and mixed forests, as well as tropical and subtropical moist broadleaf forests. In temperate regions, soil moisture is primarily controlled by precipitation amount due to moderate temperatures and limited rainfall, making substantial precipitation inputs essential for soil moisture replenishment. In contrast, tropical and subtropical regions experience high temperatures and intense evapotranspiration, requiring substantial

**precipitation to maintain a water balance." (Lines 403-417)**

Fig. 6 Ternary map of factors controlling soil moisture at annual scale, for the period 2000 to 2019. The bottom-left histogram in the subgraph represents the proportion of grid cells where one variable exerts strong univariate control (with a regression coefficient greater than 75% of the total sum of the three variables), suggesting that soil moisture was predominantly controlled by that specific variable."

**Minor comments:**

1) Line 489: the "Arctic amplification" appears abruptly. How are the climate pattern associated with the dependences? Why not other climate patterns?

Response: Our intention means that, the heterogeneity in global warming among different regions can affects GPP and drives precipitation-soil water feedback. However, climate pattern are not the focus of this study, which aims to explore the driving characteristics of GPP, air temperature, and ground temperature in different regions.

2) Line 517-519: not clear, need rephrase.

Response: The sentence was deleted in revised manuscript.

3) Line 747-748: need quantitative measures to support this point.

Response: Revised.

"Evapotranspiration was the dominant driver of soil moisture dynamics during the growing season, with a regression coefficient proportion greater than 75%. In contrast, precipitation volume played a more significant role in the surface and middle layer of non-growing season, with areas under strong univariate control accounting for over 40% of the total area. Additionally, the influence of precipitation frequency on soil moisture increased with latitude, the proportion of the regression coefficient averaging from 36.5% to 91.3%, highlighting a shift in controlling factors across climatic gradients." (Lines 625-632)

**4) Line 764-769: these reasoning needs quantitative support.**

Response: Thanks for your suggestion. Since this point is not the focus of our study, we have revised this sentence in the manuscript to better reflect our reasonable speculation.

"A possible explanation is the long-term variability in precipitation and temperature, which may have influenced geomorphology, vegetation structure, and soil water retention capacity." (Lines 648-650)

---

## Referee Report (RR1)

**Review of the paper A Study of the Dependence between Soil Moisture and Precipitation in different Ecoregions of the Northern Hemisphere**

Please note that I wrote my review before reading the comments from the first stage of the review process. Nevertheless, I share many of the same concerns previously raised, most importantly, the novelty of the paper is not clearly stated. While the research topic is certainly interesting and the relationships among the analyzed variables remain an open area of investigation, the manuscript fails to clearly explain how it contributes new insights or identifies novel relationships.

**Specific Comments:**

• Line 75: "Most current analyses of the relationship between soil moisture and precipitation assume a linear relationship."

This is an important statement. Could you please provide supporting references to substantiate this claim?

**• Data Sources:**

You use soil moisture data from ERA5, which also provides precipitation and nearsurface air temperature data. Why did you not use ERA5 for all variables, where available, to ensure consistency?

**• Line 207:**

"In this study, the ecoregion boundaries rather than Köppen climate zones were used to investigate the spatial patterns of precipitation—soil moisture feedbacks." If you are not using Köppen climate zones, why are you mentioning them here? Please clarify the purpose of this comparison.

**• Line 331:**

"The percentages of grid cells exhibiting negative dependence at these depths were 19.2%, 0.7%, and 2.3%, respectively."

Is 0.7% a meaningful value in this context? Also, the decrease in negative dependence with increasing depth is expected, but why does the middle layer exhibit *less* negative dependence than the deepest layer? This result seems counterintuitive. Can you provide a physical interpretation?

**• Lines 341–349:**

The variables  $\lambda U$  and  $\lambda L$  are formatted inconsistently. Please correct the notation. Additionally, you describe their behavior without providing any interpretation of what these patterns mean physically or statistically.

**• Monthly vs. Yearly Maps:**

How do you interpret the differences between the maps based on monthly data and those based on yearly data? This comparison is presented, but not adequately discussed.

**• Seasonality:**

Time series of climatological data are typically affected by seasonal cycles. However, you do not seem to have removed seasonality in your analysis. This is a standard procedure when applying copula models and tail dependence metrics. See, for instance:

- Neumeyer, N., Omelka, M., & Hudecová, Š. (2019). A copula approach for dependence modeling in multivariate nonparametric time series. Journal of Multivariate Analysis, 171, 139–162.
- o Durante, F., Fuchs, S., & Pappadà, R. (2025). *Clustering of compound events based on multivariate comonotonicity*. Spatial Statistics, 66, 100881.

You acknowledge the presence of seasonality, so why not attempt to deseasonalize the data and compare the results? This would add robustness to your analysis.

**• Line 379:**

"The results from ridge regression revealed more distinct patterns at the seasonal scale compared to the monthly scale (Fig. 5)."

This observation further highlights the influence of seasonality. It reinforces the concern that the analysis may benefit significantly from seasonal adjustment across the board.

**• Figure 7 and Clustering Approach:**

How are you performing clustering? In Figure 7, clusters appear to overlap. This suggests the use of fuzzy clustering, yet you previously stated (line 319) that K-means was used, which does not allow for overlapping clusters. Please clarify and correct any inconsistencies in your methodology description.

**• Section 4.3 – Data Reliability:**

This section feels largely redundant with Section 2.1. Consider consolidating or cross-referencing to avoid repetition.

**• Figure Quality:**

Some figures still appear to be of low resolution or poor quality. Please ensure all figures are adequately rendered and suitable for publication.

---

## Author Response (AR2)

Journal: Hydrology and Earth System Sciences

Title: A Study of the Dependence between Soil Moisture and Precipitation in different

Ecoregions of the Northern Hemisphere

Author(s): Shouye Xue; Guocan Wu

**Manuscript No.:** egusphere-2025-762

We highly appreciate the editor and anonymous reviewers for the very helpful and insightful comments that lead to the significant improvement this manuscript. We have checked our work carefully according to these comments and made the requested changes. In the revised version, we emphasized the new insights, improved the clarity and added more physical interpretation about the results. Figure S2 was reproduced and all figures were provided with high qualities.

Below we indicate the comments and use blue font for our responses. The corresponding revised texts are also used blue font in the revised version of our manuscript.

**Anonymous Referee #1**

The authors have made notable efforts to revise the manuscript. Most of my concerns have been addressed, and the overall clarity of the paper have improved. However, a few issues still require attention before the manuscript can be considered for publication:

1. While the authors state that lags range from 0 to 12 months, Figure S2 shows negative lags. The inconsistency needs to be addressed.

Response: Thank you for your comment. The positive values in the initial version of Figure S2 indicate that soil moisture lags behind precipitation, while negative values indicate that precipitation lags behind soil moisture. To ensure consistency with the manuscript, the negative lags were converted to positive ones by adding 12 months, which represent soil moisture lags behind precipitation in the following year. Then, Figure S2 was reproduced with lags ranging from 0 to 12 months.

Fig. S2 The estimated number of lagged month for each grid in the Kendall's tau correlation.

2. The quality of the figures remains suboptimal, especially Fig. 7 and 8.

Response: Thank you for your comment. In the revised manuscript, we have provided all figures with high qualities.

**Anonymous Referee #3**

Please note that I wrote my review before reading the comments from the first stage of the review process. Nevertheless, I share many of the same concerns previously raised, most importantly, the novelty of the paper is not clearly stated. While the research topic is certainly interesting and the relationships among the analyzed variables remain an open area of investigation, the manuscript fails to clearly explain how it contributes new insights or identifies novel relationships.

Response: Thank you for your thorough review and valuable comments. This study focused on the coupled precipitation—soil moisture system and the nonlinear dependence between precipitation and soil moisture was investigated, different from the traditional linear methods. The underlying drivers in the dependencies between soil moisture and precipitation across different regions were revealed. The results provide new insights into the spatial variability of precipitation—soil moisture dependencies and can contribute to a better understanding of how climatic and vegetation factors jointly shape soil water dynamics.

Following your comments, we have rephrased the related texts more clearly, corrected some writing issues and provided all figures with high qualities.

**Specific Comments:**

• Line 75: "Most current analyses of the relationship between soil moisture and precipitation assume a linear relationship." This is an important statement. Could you please provide supporting references to substantiate this claim?

Response: Thanks, we have added several references to support the statement.

"Most current analyses of the relationship between soil moisture and precipitation assume a linear relationship (Sehler et al., 2019; Yang et al., 2018)."

Sehler, R., J. Li, J.T. Reager, and H. Ye. 2019. Investigating Relationship Between Soil Moisture and Precipitation Globally Using Remote Sensing Observations. Journal of Contemporary Water Research & Education. 168:106-118.

Yang, L., G. Sun, L. Zhi, and J. Zhao. 2018. Negative soil moisture-precipitation feedback in dry and wet regions. Scientific Reports. 8:4026.

**• Data Sources:**

You use soil moisture data from ERA5, which also provides precipitation and near surface air temperature data. Why did you not use ERA5 for all variables, where available, to ensure consistency?

Response: Thank you for your valuable comment. In this study, we tried to use different datasets based on their observational foundations. Specifically, the CRU TS, ESA CCI, and GPCP datasets are all based on ground-based and/or satellite observations, whereas ERA5-Land is a model-derived reanalysis product. Although ERA5 does offer a wide range of meteorological variables, it can introduce model uncertainties. Therefore, the datasets used in this study have independent source, which can avoid the potential false relationships between soil moisture and precipitation that may be caused by the same model architecture and input parameters. We have added these explanations in the *Data Reliability* section of the revised version. (Lines 608-615)

**• Line 207:**

"In this study, the ecoregion boundaries rather than Köppen climate zones were used to investigate the spatial patterns of precipitation—soil moisture feedbacks." If you are not using Köppen climate zones, why are you mentioning them here? Please clarify the purpose of this comparison.

Response: Thank you for your comment. The Köppen climate zone was specifically mentioned to address the reviewer's comment regarding the regional division. To avoid confusion, we have revised the texts as follows.

"Although the Köppen climate classification provides a standardized framework based on temperature and precipitation, it may perform not well in accounting for critical biophysical factors, particularly for vegetation. Alternatively, the ecoregion divisions integrate both climatic and ecological factors, offering a more comprehensive understanding of the spatial heterogeneity in vegetation types and hydrological processes (Gerken et al., 2019; Olson et al., 2001). This makes it particularly advantageous for studying land—atmosphere interactions, since vegetation

plays a central role in regulating energy and water fluxes. Therefore, this study adopts ecoregion boundaries to better capture the vegetation related variability in precipitation—soil moisture relationship." (Line 207-216)

Gerken, T., B.L. Ruddell, R. Yu, P.C. Stoy, and D.T. Drewry. 2019. Robust observations of land-to-atmosphere feedbacks using the information flows of FLUXNET. npj Climate and Atmospheric Science. 2:37.

Olson, D.M., E. Dinerstein, E.D. Wikramanayake, N.D. Burgess, G.V.N. Powell, E.C. Underwood, J.A. D'Amico, I. Itoua, H.E. Strand, J.C. Morrison, C.J. Loucks, T.F. Allnutt, T.H. Ricketts, Y. Kura, J.F. Lamoreux, W.W. Wettengel, P. Hedao, and K.R. Kassem. 2001. Terrestrial Ecoregions of the World: A New Map of Life on Earth: A new global map of terrestrial ecoregions provides an innovative tool for conserving biodiversity. BioScience. 51:933-938.

**• Line 331:**

"The percentages of grid cells exhibiting negative dependence at these depths were 19.2%, 0.7%, and 2.3%, respectively." Is 0.7% a meaningful value in this context? Also, the decrease in negative dependence with increasing depth is expected, but why does the middle layer exhibit less negative dependence than the deepest layer? This result seems counterintuitive. Can you provide a physical interpretation?

Response: Thank you for your comment. While a general decline in negative dependence with depth is expected due to reduced soil—atmosphere coupling, the notably low percentage in the middle layer may reflect the transitional role of this zone in the soil profile. The middle soil layer, generally between 7 and 28 cm depth in ERA5-Land, commonly corresponds to the main root zone of ecosystems like temperate deciduous forests, perennial grasslands, and shrublands., The plant water uptake is relatively consistent within this zone, resulting in low variability of soil moisture. This could weaken the precipitation-soil moisture feedback signal, leading to a few grid cells with significant negative dependence (Thompson et al., 2010). In contrast, the deep soil layers may retain some long-term memory of moisture deficits, especially under prolonged dry conditions, which could contribute to stronger negative dependence than in the more buffered middle layer.

We have added these interpretations in the Discussion section. (Lines 505-512)

Thompson, S.E., C.J. Harman, P. Heine, and G.G. Katul. 2010. Vegetation-infiltration relationships across climatic and soil type gradients. Journal of Geophysical Research: Biogeosciences. 115.

**• Lines 341–349:**

The variables  $\lambda U$  and  $\lambda L$  are formatted inconsistently. Please correct the notation. Additionally, you describe their behavior without providing any interpretation of what these patterns mean physically or statistically.

Response: Thanks, we have revised all formatted error, and added discussion to clarify the significance of the observed patterns.

"This decreasing trend likely reflects the weakening of extreme precipitation—soil moisture coupling in deeper soil layers, except for arid regions where vegetation is sparse or absent." (Line 354-356)

"It revealed that the negative correlation was kept between precipitation-soil moisture in long-term scale over arid regions." (Line 361-362)

**• Monthly vs. Yearly Maps:**

How do you interpret the differences between the maps based on monthly data and those based on yearly data? This comparison is presented, but not adequately discussed.

Response: Thank you for your comment. To improve clarity, we have revised the manuscript to explicitly emphasize this contrast and its physical interpretation in the section 4.1 as follows.

"Different from monthly scale, the negative dependence at annual scale is primarily generated in regions such as deserts, xeric shrublands, montane grasslands and shrublands. These ecosystems are specifically characterized by arid conditions, and particularly sensitive to environmental changes, making them much responsive to long-term climatic variability." (Line 523-527)

**• Seasonality:**

Time series of climatological data are typically affected by seasonal cycles.

However, you do not seem to have removed seasonality in your analysis. This is a standard procedure when applying copula models and tail dependence metrics. See, for instance:

Neumeyer, N., Omelka, M., & Hudecov., Š. (2019). A copula approach for dependence modeling in multivariate nonparametric time series. Journal of Multivariate Analysis, 171, 139–162.

Durante, F., Fuchs, S., & Pappad., R. (2025). Clustering of compound events based on multivariate comonotonicity. Spatial Statistics, 66, 100881.

You acknowledge the presence of seasonality, so why not attempt to de-seasonalize the data and compare the results? This would add robustness to your analysis.

Response: Thank you for your valuable comment. We agree that removing seasonal cycles prior to dependence analysis could help isolate the intrinsic relationship between precipitation and soil moisture, and improve the statistical robustness. We have revised the manuscript to state this limitation and added more explanations.

"The copula method can access the dependence between different time series, after removing influences of the conditional means and variances as well as marginal distributions (Neumeyer et al. 2019; Durante et al. 2025). In this study, although precipitation—soil moisture dependence was assessed across different time scales, the monthly series were not de-seasonalized. As a result, the residual seasonal signals may influence short-term dependence structures. This limitation will be addressed in future work through seasonal adjustment." (Line 619-625)

**• Line 379:**

"The results from ridge regression revealed more distinct patterns at the seasonal scale compared to the monthly scale (Fig. 5)."

This observation further highlights the influence of seasonality. It reinforces the concern that the analysis may benefit significantly from seasonal adjustment across the board.

Response: Thank you for raising this point. While our analysis includes results at the

seasonal scale, it is important to clarify that the dependence measurements were computed for each season separately on an annual basis (e.g., spring 2000, spring 2001, ..., spring 2019), rather than aggregating all seasons together or combining all months across years. Therefore, each seasonal estimate is based on data with a one-year interval, which reduces the influence of intra-annual periodicity and helps to mitigate the effect of strong seasonal cycles.

Therefore the influence of seasonality on the ridge regression results is not overstated, and that the generated dependence patterns can reflect the differences in precipitation—soil moisture coupling behavior across different time scales. We have clarified this point in the revised manuscript to avoid misunderstanding.

**• Figure 7 and Clustering Approach:**

How are you performing clustering? In Figure 7, clusters appear to overlap. This suggests the use of fuzzy clustering, yet you previously stated (line 319) that K-means was used, which does not allow for overlapping clusters. Please clarify and correct any inconsistencies in your methodology description.

Response: Thank you for your comment. The K-means clustering was used in this study, which assigns each data point to a single, non-overlapping cluster. The appearance of overlapping clusters in Figure 7 is likely due to visual effects caused by spatial aggregation or the dominance of a single cluster in some regions. We have clarified this point in the revised manuscript and figure caption to avoid misunderstanding.

**• Section 4.3 – Data Reliability:**

This section feels largely redundant with Section 2.1. Consider consolidating or cross-referencing to avoid repetition.

Response: Thanks for your advices. We have revised section 4.3 based your comment as follows. (Lines 608-631)

"In this study, multiple observational datasets were employed to reduce model-driven uncertainty and enhance data reliability. CRU TS, ESA CCI, and GPCP

were selected due to their direct reliance on ground-based or satellite observations, in contrast to the reanalysismodel-based ERA5-Land product. Although ERA5 does offer a wide range of meteorological variables, it can introduce model uncertainties. Therefore, the datasets used in this study have independent source, which can avoid the potential false relationships between soil moisture and precipitation that may be caused by the same model architecture and input parameters. To investigate spatial heterogeneity, all data were spatially aggregated by ecoregion boundaries from the Conservation Biology Institute. These boundaries may introduce regional biases, which should be considered when interpreting the results.

The copula method can access the dependence between different time series, after removing influences of the conditional means and variances as well as marginal distributions (Durante et al., 2025; Neumeyer et al., 2019). In this study, although precipitation—soil moisture dependence was assessed across different time scales, the monthly series were not de-seasonalized. As a result, the residual seasonal signals may influence short-term dependence structures. This limitation will be addressed in future work through seasonal adjustment. In the Bayesian modeling, GPP, LST, and air temperature were examined as drivers of negative dependence. Evapotranspiration was excluded due to its dependence on both soil moisture and temperature. We acknowledge that additional factors—such as wind, topography, and soil physical properties—may also modulate precipitation—soil moisture coupling but were not in the scope of this analysis. Future research incorporating these variables would provide a more comprehensive understanding of the underlying mechanisms."

**• Figure Quality:**

Some figures still appear to be of low resolution or poor quality. Please ensure all figures are adequately rendered and suitable for publication

Response: Thank you for your comment. In the revised manuscript, we have provided all figures with high qualities.

---

## Author Response (AR3)

Journal: Hydrology and Earth System Sciences

Title: A Study of the Dependence between Soil Moisture and Precipitation in different

Ecoregions of the Northern Hemisphere

Author(s): Shouye Xue; Guocan Wu

Manuscript No.: egusphere-2025-762

We highly appreciate the editor for the very helpful and insightful comments that lead to the significant improvement this manuscript. We have checked our work carefully according to these comments and made the requested changes. In this revised version, we emphasized the novelty of this study, and clearly explained how it contributes the new insights.

Below we indicate the comments and use blue font for our responses. The corresponding revised texts are also used blue font in the revised version of our manuscript.

**Comments from Editor**

Your revised manuscript received substantially good feedbacks from the Referees. Although, could be further improved by taking into account the comments raised by the Reviewers, according to the statements in your reply. Please proceed according to your response and submit a final revised version of your work.

Response: Thank you for your recommendation and comments. We emphasized the novelty of this study, and clearly explained how it contributes the new insights, based on the version of "egusphere-2025-762-manuscript-version4". Some writing typos were also corrected.

**Lines 18-20:**

The relationship between soil moisture and precipitation was found to be nonlinear and negative in Northern Hemisphere ecosystems.

**Lines 22-24:**

This study quantified the spatiotemporal distribution of the nonlinear dependence of soil moisture to precipitation, and identify the dominant factors in different ecoregions to explore the driving mechanisms and regional patterns.

**Lines 78-81:**

Existing studies have not fully addressed some issues in the nonlinear dependence of soil moisture to precipitation, including the heterogeneity in different ecoregions and soil layers, as well as inadequate identification of tail dependence.

**Lines 108-110:**

In particular, although the negative dependence has been reported, its dominant drivers and their relative contributions across different timescales and soil layers still remain unclear.

**Lines 118-129:**

This study targets the nonlinear dependence of soil moisture to precipitation across Northern Hemisphere at monthly and annual scales from 2000 to 2019. A copula function was applied to describe the joint distribution of precipitation and soil moisture. It can capture the asymmetric and tail-dependent relationship, as well as the varying influences of precipitation volume, frequency, and evapotranspiration on soil moisture at monthly and seasonal scales. A Bayesian attribution framework involved gross primary productivity (GPP), land surface temperature (LST), and near-surface air temperature (Ta) were selected to identify the key driving factors, since the dependence between precipitation and soil moisture is influenced by factors such as vegetation growth, temperature, and soil properties. The driving factors and regional characteristics of the negative correlation observed between precipitation and soil moisture in different ecoregions were also compared.

**Lines 629-631:**

Bayesian models were used to analyze the driving factors and relative contribution in the dependence of soil moisture to precipitation in different time scales and ecoregions of the Northern Hemisphere.

Again, we thanks for your recommendation and comments.

---

## Author Response (AR4)

Journal: Hydrology and Earth System Sciences

Title: A Study of the Dependence between Soil Moisture and Precipitation in different

Ecoregions of the Northern Hemisphere

Author(s): Shouye Xue; Guocan Wu

Manuscript No.: egusphere-2025-762

We highly appreciate the editor for the very helpful and insightful comments that lead to the significant improvement this manuscript. We have checked our work carefully according to these comments and made the requested changes. In this revised version, we reorganized the conclusions to emphasize the new findings.

Below we indicate the comments and use blue font for our responses. The corresponding revised texts are also used blue font in the revised version of our manuscript.

**Comments from Editor**

Please, let us know what are YOUR original conclusions. About statements in the conclusions: Do you confirm/extend a well-known observation? Provide citation..Is this a completely new concept? ...state it clearly.

Response: Thank you for your comments. We revised the statements in the conclusions, by providing citations for the well-known parts and emphasizing the new findings, based on the version of "egusphere-2025-762-manuscript-version5".

**Examples of statements:**

"Precipitation volume predominantly controlled soil moisture in the Boreal forest/taiga, temperate grasslands, savannas, and shrublands, while precipitation frequency primarily controlled soil moisture in the high-latitude regions of the Northern Hemisphere."

Response: "Our studies have new insight for the dependence of soil moisture to precipitation varying in different ecoregions. We concluded that, precipitation volume predominantly controlled soil moisture in the Boreal forest/taiga, temperate grasslands, savannas, and shrublands, while precipitation frequency primarily controlled soil moisture in the high-latitude regions of the Northern Hemisphere." (Lines 634-638)

"Evapotranspiration was the dominant driver of soil moisture dynamics during the growing season"

Response: "While evapotranspiration is known to dominate soil moisture dynamics during the growing season (Kozii et al., 2020), this study quantified that this dominance are with regression coefficients more than 75% of the total sum of the three covariates." (Lines 639-642)

Kozii, N., K. Haahti, P. Tor-ngern, J. Chi, E.M. Hasselquist, H. Laudon, S. Launiainen, R. Oren, M. Peichl, J. Wallerman, and N.J. Hasselquist. 2020. Partitioning growing season water balance within a forested boreal catchment using sap flux, eddy covariance, and a process-based model. *Hydrol. Earth Syst. Sci.* 24:2999-3014.

"In regions such as temperate grasslands, savannas, shrublands, deserts, xeric shrublands, and tundra, negative dependencies between precipitation and soil moisture, driven by LST and Ta:GPP interactions, were observed..."

Response: "For the factor driving the dependence of soil moisture to precipitation, this study found that the negative dependences were distributed across temperate

grasslands, savannas, shrublands, deserts, xeric shrublands, and tundra, primarily driven by LST and Ta:GPP interactions." (Lines 648-651)

"At the annual scale, the area of negative dependence increased with soil depth, with the most pronounced negative dependencies occurring in the montane grasslands and shrublands region. In this region, negative dependencies at all three soil depths were driven by the GPP:LST interaction..."

Response: "At the annual scale, the negative dependences were mainly in the montane grasslands and shrublands region (Wei et al., 2008). This study further revealed that this negative dependence increased with soil depth, and were driven by the GPP:LST interaction across all three soil layers." (Lines 661-664)

Wei, J., R.E. Dickinson, and H. Chen. 2008. A Negative Soil Moisture–Precipitation Relationship and Its Causes. *Journal of Hydrometeorology*. 9:1364-1376.

The reorganized conclusion section is as follows.

This study explored the dependence relationships between precipitation and soil moisture at depths of 0 to 7 cm, 7 to 28 cm, and 28 to 100 cm from 2000 to 2019, by examining the control effect of precipitation volume, precipitation frequency, and evapotranspiration on soil moisture. Bayesian models were used to analyze the driving factors and relative contribution in the dependence of soil moisture to precipitation in different time scales and ecoregions of the Northern Hemisphere. The results showed that, the negative dependence proportion reached 19.2%, 0.7%, and 2.3% at monthly scale, while it was 3.0%, 4.0%, and 8.6% at annual scale, respectively, for the three soil layers. Our studies have new insight for the dependence of soil moisture to precipitation varying in different ecoregions. We concluded that, precipitation volume predominantly controlled soil moisture in the Boreal forest/taiga, temperate grasslands, savannas, and shrublands, while precipitation frequency primarily controlled soil moisture in the high-latitude regions of the Northern Hemisphere. The combined influence of evapotranspiration and precipitation exhibited clear seasonal patterns. While evapotranspiration is known to dominate soil moisture dynamics during the growing season (Kozii et al., 2020), this study quantified that this dominance are with regression coefficients more than 75% of the total sum of the three covariates. In contrast, precipitation volume played a more significant role in the surface and middle layer of non-growing season, with areas under strong univariate control accounting for over 40% of the total area. Additionally, the influence of precipitation frequency on soil moisture increased with latitude, the proportion of the regression coefficient averaging from 36.5% to 91.3%, highlighting a shift in controlling factors across climatic gradients.

For the factor driving the dependence of soil moisture to precipitation, this study found that the negative dependences were distributed across temperate grasslands, savannas, shrublands, deserts, xeric shrublands, and tundra, primarily driven by LST and Ta:GPP interactions. These negative dependences were mainly attributed to the seasonality of precipitation in arid and semi-arid areas and the freeze—thaw processes in the soil, which hinder effective moisture replenishment, especially during winter when soil freezing prevents rainwater infiltration. In the intermediate and deep soil layers, negative dependences were primarily driven by single variables, whereas positive dependences resulted from multivariate interactions, likely due to the lack of compensatory mechanisms when a single variable dominated, or the enhancement of ecosystem feedbacks when both GPP and LST interacted. Additionally, when the ecosystem is simultaneously driven by GPP and LST, greater resilience may be exhibited.

At the annual scale, the negative dependences were mainly in the montane grasslands and shrublands region (Wei et al., 2008). This study further revealed that this negative dependence increased with soil depth, and were driven by the GPP:LST interaction across all three soil layers. A possible explanation is the long-term variability in precipitation and temperature, which may have influenced geomorphology, vegetation structure, and soil water retention capacity.

Again, we thank for your recommendation and valuable comments.